# Advancement in Soft Hydrogel Grippers: Comprehensive Insights into Materials, Fabrication Strategies, Grasping Mechanism, and Applications

**DOI:** 10.3390/biomimetics9100585

**Published:** 2024-09-27

**Authors:** Xiaoxiao Dong, Chen Wang, Haoxin Song, Jinqiang Shao, Guiyao Lan, Jiaming Zhang, Xiangkun Li, Ming Li

**Affiliations:** 1College of Mechanical Engineering, Liaoning Petrochemical University, Fushun 113001, China; waen2023@163.com (C.W.); songhaoxin6@gmail.com (H.S.); sjq20011030@163.com (J.S.); 18742172064@163.com (G.L.); zhang2580213819@163.com (J.Z.); 18382023580@163.com (X.L.); 2Center for Advanced Structural Ceramics, Department of Materials, Imperial College London, London SW7 2AZ, UK

**Keywords:** hydrogel gripper, hydrogel materials, stimuli-responsive mechanism, grab and release objects

## Abstract

Soft hydrogel grippers have attracted considerable attention due to their flexible/elastic bodies, stimuli-responsive grasping and releasing capacity, and novel applications in specific task fields. To create soft hydrogel grippers with robust grasping of various types of objects, high load capability, fast grab response, and long-time service life, researchers delve deeper into hydrogel materials, fabrication strategies, and underlying actuation mechanisms. This article provides a systematic overview of hydrogel materials used in soft grippers, focusing on materials composition, chemical functional groups, and characteristics and the strategies for integrating these responsive hydrogel materials into soft grippers, including one-step polymerization, additive manufacturing, and structural modification are reviewed in detail. Moreover, ongoing research about actuating mechanisms (e.g., thermal/electrical/magnetic/chemical) and grasping applications of soft hydrogel grippers is summarized. Some remaining challenges and future perspectives in soft hydrogel grippers are also provided. This work highlights the recent advances of soft hydrogel grippers, which provides useful insights into the development of the new generation of functional soft hydrogel grippers.

## 1. Introduction

Soft grippers, as a new generation of mechanical gripper, have the ability to recognize, grasp, and release objects and interact with the environment [1,2,3,4,5]. The research on soft grippers involves contributions from many organisms, including worms [6,7,8], chameleons [9], snakes [10], geckos [6], frogs [11], fish [12,13,14], and octopuses [15,16]. Compared with conventional rigid mechanical grippers, soft grippers have advantages including low mechanical impedance and relatively high compliance in a safe state, which make them applicable in healthcare, soft robotics, aviation, the military, etc. [17,18,19,20].

In-depth exploration of materials and manufacturing holds the potential to expedite the advancement of soft grippers with desirable characteristics [21,22,23]. From the perspective of materials, hydrogels have attracted scientists’ attention due to their three-dimensional network structure, whose volume changes as swelling and deswelling occur in response to external stimuli, such as temperature, pH, and electromagnetism. Easy fabrication, low cost, excellent mechanical properties, and fast response of hydrogels are also outstanding reasons to choose them [24,25,26]. It should be emphasized that the state of response is intimately associated with the monomers of hydrogel polymerization, crosslinking agents, and the polymerization process [27,28,29,30]. And the response-driven characteristics of hydrogels endow them with unique advantages in the preparation of grippers [31,32,33]. Certain hydrogels do not possess remarkable response-driven characteristics. However, this sort of hydrogel can be the preferred material for the manufacturing of soft grippers on account of their rich functional group structure in conjunction with other materials [21,34,35,36]. This particular type of soft gripper in which hydrogels are involved is defined as a soft hydrogel gripper in this paper. Compared with soft responsive materials often used in manufacturing fixtures, such as shape memory polymers, dielectric elastomers, and liquid-crystal elastomers, hydrogels have advantages. They have good biocompatibility, excellent softness, and diverse environmental responsiveness. However, they also have limitations. Their mechanical properties are low, and they are prone to damage under large external forces. Their response speed is slow, and it is difficult to adjust in scenarios requiring rapid response. Moreover, they contain a large amount of water, and their structure and performance change significantly or even become inactivated when dried. In contrast, other materials have advantages in terms of mechanical properties, response speed, and stable performance in dry environments.

Besides the efforts made in the aspect of materials, research into the fundamental manufacturing methods in soft grippers has spurred advancements in excellent response effect and mechanical properties. Nowadays, laser cutting, injection molding, and 3D printing are rather common methods to prepare soft hydrogel grippers [37,38]. The choice of soft hydrogel gripper manufacturing methods depends on hydrogel types, for example, the swelling characteristics and dehydration shrinkage state of hydrogels limit the materials to combine with them [39,40]. Generally speaking, hydrogels combine with polymer materials to achieve environmental-stimuli-responsive actuation to further fabricate soft hydrogel grippers. Not all materials possess the direct compatibility feature, therefore, avoiding incompatibility or a swelling phenomenon during the combination process of hydrogel with other materials is a key science problem. How to maintain the external response and special inner characteristics of hydrogel is an essential point to consider as well. To counteract these, the research about heterogeneous interface combination from physical and chemical aspects plays a vital role. The H-bonds, metal coordination, π–π stacking electrostatic interaction, hydrophobic interaction, and host–guest interaction from a physical aspect and free radical polymerization, Michael addition, borate ester bonds, Schiff base bonds, disulfide bonds, and Diels–Alder reaction from chemical aspect are common methods to achieve heterogeneous interface combinations of hydrogel and other materials [40,41,42,43,44]. As well as various characteristics of surface or interface combinations, it is worth mentioning that doping multifunctional particles or elements into hydrogel to achieve environmental response actuation is also a good way [41,42,43,44]. For a better grasp and release state, the interaction and response of hydrogel matrix and functional particles need further study. As a whole, to fabricate ideal performance stimuli-responsive hydrogel grippers, there are two main categories: (i) hydrogel networks responding to external stimuli directly, (ii) stimuli-responsive particle/elements serving as fillers introduced into the hydrogel system.

In addition, it is important to highlight that optimization of the structure and materials ratio directly affects properties of soft hydrogel grippers [45,46,47,48]. Therefore, optimization experiments are usually an indispensable link in the research of a high-quality soft hydrogel gripper. The introduction of finite analysis and machine learning also provides effective prediction and assistance for the design and preparation of hydrogel grippers. Notably, recent advancements have been made in the development of soft hydrogel grippers capable of high load capability, long service life, and fast response, particularly for actual application in extreme environments to achieve controlled grasp and release of objects.

In the early stages of commercialization and industrialization, hydrogel grippers were mostly in the laboratory research phase. Researchers explored their basic properties and attempted to apply them to gripper design. Although they recognized the potential advantages, it was difficult to achieve commercial applications due to technological limitations. In order to meet production requirements, researchers made efforts in two aspects: material improvement (adding nanomaterials, etc. to optimize performance and adopting new methods to increase speed) and prototype development (using optimized materials to develop simple prototypes and having initial feasibility in biomedical research).

This review focuses on stimuli-responsive soft hydrogel grippers. It aims to explore the material composition and characteristics of stimuli-responsive hydrogels and also to explore the relevant efforts in combining hydrogels with other materials to fabricate various such grippers (Figure 1). Specifically, the Introduction expounds the research background of soft grippers and their application potential in multiple fields. The materials part elaborates on the molecular structures, functional groups, and characteristics of common hydrogels and their influence on the actuation of hydrogel grippers. The fabrication strategies part includes one-step synthesis of direct polymerization and fabrication additives, as well as structural modification strategies such as bilayer, patterned, multilayer, oriented, and gradient structures and also includes doping strategies, etc. The stimuli-responsive hydrogel grippers part discusses hydrogel grippers under external stimuli-responsive mechanisms such as thermal, electrical/magnetic, and chemical stimuli, explores other response types, and expounds on multiresponsive hydrogel grippers. Reviewing the latest progress in this field to introduce the applications of soft hydrogel grippers is also a key content of this review. Finally, the conclusion and outlook part summarizes the research progress, points out the existing problems, and presents an outlook for future development.

## 2. Materials of the Hydrogel Grippers

To achieve excellent performance of hydrogel grippers, the selection of hydrogel plays an essential role. In this section, we delve into various common applied hydrogels tailored with PNIPAM, PAA, PVA, PEGDA, and P(MAAM-*co*-MAA) which are relevant to hydrogel grippers (Figure 2). We introduce the molecular structure, functional groups, and characteristics of hydrogel in detail. These descriptions provide valuable insights into the mechanisms of hydrogel grippers’ actuation.

### 2.1. PNIPAM

Poly N-isopropylacrylamide (PNIPAM) has temperature-sensitive properties due to the simultaneous presence of hydrophilic amide groups (-CONH-) and hydrophobic isopropyl groups [-CH(CH_3_)_2_-] on its molecular chain (Figure 2a) [49,50,51]. That is to say, the lower critical solution temperature (LCST) of PNIPAM is 32 degrees Celsius. At low temperatures, the interaction between PNIPAM and water is mainly the hydrogen bonding between amide groups and water molecules. The water molecules around the macromolecular chain will form a hydrogen-bonded, highly ordered solvation layer and make the polymer show an extended coil structure. As the temperature rises, the interaction parameter between PNIPAM and water changes abruptly, and some hydrogen bonds are destroyed. The solvation layer of the hydrophobic part of the macromolecular chain is then destroyed. At this time, the polymer changes from a loose coil structure to a compact colloidal particle structure, thereby generating temperature sensitivity.

This property enables the preparation of grippers that respond to external temperature. This is because the morphology of PNIPAM can be altered by controlling the temperature. By precisely controlling the temperature change, it can quickly and effectively respond to and execute gripping actions in accordance with changes in external temperature [52,53].

### 2.2. PAA

Polyacrylic acid (PAA) is an important water-soluble polymer. The PAA molecular chain contains a large number of carboxyl (-COOH) functional groups, and these carboxyl groups are able to ionize hydrogen ions (H^+^) in aqueous solution to form negatively charged polyanions (Figure 2b) [54,55,56]. The degree of ionization of the carboxyl groups is affected by the PH of the solution and the degree of ionization and charge state of the PAA changes with the change in PH. This PH sensitivity results in the solubility (i.e., the ability to absorb water and swell) of the PAA hydrogel varying with solution PH. In an alkaline environment, the carboxyl groups in the hydrogel molecules ionize, the intermolecular electrostatic repulsion is enhanced, and the hydrogel undergoes swelling; in an acidic environment, the carboxyl groups in the hydrogel molecules cannot ionize, and the molecules are connected by hydrogen bonding, which results in the shrinkage of the hydrogel.

This PH-sensitive characteristic can be utilized to fabricate grippers that respond to the external environment. Precise control of the PH value can enable the robotic hand to respond quickly and effectively to changes in the external environment and perform grasping and releasing actions [57,58].

### 2.3. PVA

Polyvinyl alcohol (PVA) is a polymer with good chemical stability and biological safety [59]. The main chain of PVA molecules is a carbon chain, and the side chains contain a large number of hydroxyl groups [60,61,62]. The hydrophilicity of PVA mainly comes from the hydroxyl groups in its molecules, and the hydrophobicity comes from the acetyl groups (Figure 2c). These hydroxyl groups can form hydrogen bonds with water molecules due to their high activity, providing PVA with good hydrophilic properties, while the acetyl groups contribute to the hydrophobicity of PVA. This dual nature enables PVA to exhibit unique solubility and surface-active behavior in aqueous solutions. In addition, temperature has a significant impact on the behavior and interaction of PVA molecular chains. Due to the presence of hydrophilic and hydrophobic groups in its molecular structure, these groups are sensitive to temperature changes. At low temperatures, the hydrogen bond interaction between PVA molecular chains is relatively stable, and the hydrogel can maintain a swollen state. However, when the temperature rises to a certain extent, the hydrogen bond interaction weakens, and the hydrophobic interaction of the PVA molecular chain begins to dominate, resulting in the contraction of the hydrogel, thereby exhibiting temperature sensitivity.

By adjusting the temperature to control the hydrophilicity and hydrophobicity of PVA molecules, the state of the hydrogel fixture can be regulated, thereby achieving rapid and effective grasping and releasing actions [63,64].

### 2.4. PEGDA

Polyethylene glycol diacrylate (PEGDA) is a material composed of polyethylene glycol (PEG) segments. The PEG segments have a linear structure without branches, and this structural characteristic endows them with extraordinary flexibility and solubility [65]. It is precisely because of this flexibility and solubility that PEGDA molecules can more easily undergo conformational changes and interactions under light [66,67,68]. Moreover, it also affects the compatibility of PEGDA with other components (such as photoinitiators, additives, etc.), thereby further affecting the efficiency and effect of the photosensitive reaction. PEGDA contains unsaturated double bonds, and the molecular structure of PEGDA contains acrylate groups, which carry unsaturated double bonds (C=C) (Figure 2d). In the presence of a photoinitiator, the unsaturated double bonds easily absorb light of specific wavelengths. For example, under the irradiation of ultraviolet light, these double bonds will be activated and undergo a chemical reaction. The existence of this double bond enables PEGDA to participate in the photoinitiated polymerization reaction, which is the key structural basis for its photosensitivity.

By taking advantage of the photosensitive characteristic of PEGDA, which responds to specific lighting conditions, a fixture that can perform grasping and releasing actions according to light can be designed [69].

### 2.5. P(MAAM-co-MAA)

The molecular structure of methacrylamide-*co*-methacrylic acid P(MAAM-*co*-MAA) includes methacrylamide and methacrylic acid (Figure 2e,f) [70]. In an acidic environment, carboxylic acid groups accept hydrogen ions (protonation) and exist in the form of -COOH. In an alkaline environment, carboxylic acid groups lose hydrogen ions (deprotonation) and form carboxylate ions (-COO-). This process of protonation and deprotonation changes the charge distribution of the copolymer and then affects its expansion and contraction behavior. The change in charge state caused by PH change will also affect the configuration of the copolymer molecular chain. When protonated, the molecular chain is relatively curly. However, when deprotonated, the molecular chain tends to stretch. This configuration change directly affects the volume and mechanical properties of materials.

P(MAAM-*co*-MAA) copolymer has many excellent properties because of its special molecular structure, among which the hardness and impact resistance are high, which helps it to maintain stability and durability during grasping. Due to its excellent mechanical properties, PH sensitivity, and unique swelling and crosslinking characteristics, P(MAAM-*co*-MAA) can be used to successfully prepare grippers that respond to external PH.

The materials of the hydrogel gripper play a crucial role in its performance and functionality [71,72]. Different materials offer unique properties that enable the gripper to respond to various external stimuli such as temperature, pH, and light. These materials mentioned above reflect that hydrogels generate different responses to these external stimuli. These external stimuli alter the hydrophilic and hydrophobic properties of hydrogels, thereby making them soft and fine. As a result, it is possible to achieve the grasping and releasing of objects. In conclusion, the materials employed in hydrogel grippers possess diverse characteristics. These characteristics enable the creation of grippers that can respond to various external stimuli, thus providing a broad range of applications in different fields.

## 3. Manufacturing Strategies

Due to the variations in properties among different materials, hydrogel grippers fabricated with diverse matrices take various manufacturing strategies into account. Manufacturing strategies here refer to selective synthesis mode and fabrication technology divided into one-step synthesis and structure modification. One-step synthesis, including direct polymerization and manufacturing additives, is carried out according to the targeted gripper structure. Structure modifications including bilayer construction, patterned structure, multilayer structure, oriented structure, and gradient structure are summarized in detail.

### 3.1. One-Step Synthesis

In the realm of manufacturing hydrogel grippers, an array of novel and highly efficient preparation strategies are continuously emerging. Among these, this section primarily elaborates on direct polymerization and the manufacturing additive technology. Direct polymerization, characterized by its conciseness and efficiency, can swiftly convert raw materials into hydrogel grippers with specific properties. Meanwhile, the manufacturing additive technology, leveraging its precise customization advantage, offers the possibility of achieving complex structures and personalized designs [73].

The one-step polymerization of hydrogels focuses on the direct reaction and crosslinking process initiated by monomers or polymers under specific conditions. In the direct polymerization process, the required reactants (e.g., monomers, crosslinking agents, and initiators) are usually mixed all at once. During this process, the initiator can initiate the polymerization reaction of the monomers. The crosslinking agent will drive the construction of crosslinking structures between the polymer chains at the same time, thereby directly generating the hydrogel in a single step (Figure 3a) [74]. The key core of this method lies in the precise control of reaction conditions, which covers many aspects such as temperature, pH value, reactant concentration, etc. Only by ensuring that these reaction conditions can be accurately adjusted and controlled can the reaction proceed efficiently and uniformly and thus form a hydrogel with the expected performance and structure. Direct polymerization has many advantages, such as relatively simple and convenient operation, the ability to obtain hydrogel products quickly, and reducing the errors and complexity that may be caused by multistep operations. However, it should be noted that one-step polymerization has high requirements for the control of reaction conditions, which requires a deep understanding of each related factor and the ability to make accurate judgments and precise control. When preparing the hydrogel as a whole and aiming to transform it into a gripper, external auxiliary methods like laser cutting and plastic molding with molds are frequently employed. Further, electrode assistance shows more significant advantages in this transformation process. Zhu et al. [74] developed biomimetic deformable hydrogels with complex anisotropic structures by generating distributed electric fields for the electrical orientation of nanosheets (NSs) through electrode-assisted polymerization using patterned electrodes. Anisotropic hydrogels with thickness gradients can be fabricated by utilizing interlaced pectinate electrodes that are plated on one substrate of the reaction cell. In contrast, the other substrate of the reaction cell is not equipped with electrodes. Anisotropic hydrogels with in-plane gradients can be developed by using a pair of patterned electrodes plated on the two substrates of the reaction cell (Figure 3b) [74]. Electrodes featuring an identical pattern are assembled in a face-to-face manner to eliminate the thickness gradient. Nevertheless, owing to the disparities in the arrangement of NSs between electrodes of opposite polarities and those of the same polarity, an in-plane gradient is generated. Hydrogel grippers with extensive application scopes, along with functional structures and biomimetic materials, can be developed by means of this approach.

The manufacturing processes of hydrogels mostly rely on two-dimensional (2D) synthesis methods, yet this constitutes a limitation for the in-depth development of additive manufacturing hydrogels [77]. Moreover, the common direct polymerization method usually has limitations in terms of structure. To achieve the preparation of the ideal hydrogel gripper, external assistance is needed to realize the desired structure. Consequently, additive manufacturing technology (also known as 3D printing), being a method with relatively high 3D synthesis accuracy, controllable structure, and straightforward operation, is undoubtedly an excellent option for the fabrication of 3D hydrogels. In the practice of 3D printing hydrogels, the importance of printing materials is self-evident. Ordinary hydrogels often encounter many difficulties in the printing process due to their high water content. For this reason, researchers have discovered that incorporating an appropriate amount of materials such as glycerol into the hydrogel or other methods enhances their water retention capacity. Through such modification of the materials, the printability of the hydrogel can be effectively improved. This fully indicates that the careful selection and reasonable modification of printing materials are the key to achieving high-quality 3D printing of hydrogels. Zhan et al. [75] developed a photothermally responsive hydrogel based on n-isopropylacrylamide and fabricated it through 3D printing technology based on projection microstereolithography. The researchers initially added NIPAAm into deionized water (DI) and stirred it magnetically. Subsequently, AAM and PEGDA were continuously stirred to dissolve them in the former solution (solution A). Ethyl (2,4,6-trimethylbenzoyl) phenylphosphinate (TPO-L) was dissolved in acetone (solution B), and then it was added to the previous solution. Given that TPO-L cannot be directly dissolved in water, acetone was used as a transitional solvent. After magnetic stirring, the hydrogel precursor solution was successfully prepared. Subsequently, a shape memory hydrogel structure with carbon nanoparticles was decorated on the printed structure. Through 3D printing technology based on PµSL, high-resolution 3D complex structures can be generated (Figure 3c). Thus, an ultrafast photothermal responsive shape memory hydrogel was prepared. Due to the sharp temperature change and water absorption and loss under near-infrared light irradiation, the light-driven hydrogel microrobot can achieve a rapid response time. By adjusting the light intensity and changing the irradiation position, the shape and swelling movement of the hydrogel can be easily controlled. This design provides a novel method for ultrafast photothermal responsive hydrogel microrobots working in water. With the continuous development and improvement of 3D printing hydrogel technology, the scope of research has gradually expanded to a more cutting-edge field of 4D printing hydrogels. The 4D printing hydrogel possesses numerous significant advantages. Firstly, its environmental response capability is extremely excellent and it can change its shape, performance, or function according to changes in external factors such as temperature, humidity, pH value, light, and so on. Secondly, its programmability is rather powerful. Finally, it can achieve dynamic changes in the time dimension. Hua et al. [76] used 4D printing technology to graft NIPAM onto a PVA-MA backbone, making PVA thermally responsive, and successfully prepared PVA/(PVA-MA)-g-PNIPAM hydrogels through 4D printing technology based on digital light processing (DLP), successfully manufacturing hydrogel actuators of different geometric shapes with high toughness, actuation force, and shrinkage rate, and its actuation speed is 4 times faster than that of bulk hydrogels. The researchers dissolved PVA, PVA-MA, NIPAM, and TPO-Li in water, and then exposed the mixture to ultraviolet light to achieve gelation (Figure 3d). No crosslinking agent was added during this operation to ensure that all remaining NIPAM molecules could attach to the PVA-MA skeleton after rinsing. The fabricated hydrogel appears transparent and is colored by adding a light-absorbing dye to its precursor (Figure 3d-i). The sample changed from transparent to translucent after salt precipitation treatment and remained translucent even when immersed in water, indicating the formation of nanocrystalline domains that can strongly scatter light (Figure 3d-ii). Thanks to the single-network design, the toughened PVA/(PVA-MA)-g-PNIPAM hydrogel maintained reversible thermal responsiveness in water under salt precipitation (Figure 3d-iii). The proposed design is beneficial for hydrogel manufacturing and application, providing a new method for designing hydrogel grippers with other response methods.

Overall, the one-step polymerization and the printing method have demonstrated their respective unique charm and significant potential in the preparation of hydrogel grippers. They not only provide powerful technical support for current research and applications but also open up broader space for the future development of hydrogel grippers and are expected to promote continuous new breakthroughs and innovations in this field.

### 3.2. Structure Modification

In the process of crosslinking hydrogels, the complex structural characteristics and the diverse range of hydrogel materials enable these hydrogels to be actuated in response to stimuli such as light, heat, magnetism, and chemical elements within clamping devices. This section focuses on summarizing five structural categories of hydrogel grippers: bilayer structures, patterned structures, multilayer structures, oriented structures, and gradient structures. Furthermore, it delves into the different types of composite hydrogel materials, the functions of each hydrogel layer, and the various crosslinking methods employed. This discussion aims to elucidate the latest advancements in hydrogel grippers based on structural modifications, while also offering concise and innovative perspectives for future development [78].

Bilayer hydrogels are composed of two hydrogels with different levels of water absorption and swelling, each layer playing a specific role within the overall structure due to their distinct physicochemical properties. This simple yet effective structure has been extensively studied for its ability to achieve controllable deformation (bending and buckling) through asymmetric response characteristics. Additionally, the layer-by-layer assembly allows for precise tuning of properties by adjusting the composition or thickness of each layer, further enhancing its potential applications [79,80,81,82,83]. A bilayer hydrogel is composed of two distinct layers: an upper layer and a lower layer. The upper layer is typically a hydrogel that is highly sensitive to external stimuli, such as light, temperature, pH, electric fields, or magnetic fields. Upon stimulation, this layer undergoes significant volume changes (e.g., swelling or contraction), which drive the deformation of the entire structure. Therefore, it is often referred to as the actuation layer, driving layer, or active layer. Its primary function is to sense and respond to external signals during interaction with the environment, such as light, magnetic, or electric responsiveness. These functional materials enable the hydrogel to exhibit specific actuation behaviors, such as bending under light exposure or extending in an electric field. The lower layer typically serves as a support or substrate, providing mechanical stability and structural integrity to the entire bilayer structure and is thus referred to as the passive layer or support layer. This layer does not require the same level of responsiveness to external stimuli as the upper layer, but its material properties, such as elastic modulus and crosslinking density, play a critical role in influencing the deformation pattern of the overall structure. The lower layer can also regulate deformation behavior through its synergistic interaction with the upper layer. For example, when the upper layer swells, the relative stability or differing swelling coefficients of the lower layer can lead to complex deformations in the hydrogel structure, such as bending, curling, or folding. In some cases, the lower hydrogel layer is itself insensitive to external stimuli, with its primary function being to constrain or guide the deformation of the upper layer, thereby controlling the final movement pattern or shape change. This design ensures that soft robots or actuators generate the desired deformation in specific directions rather than arbitrary ones. Thus, the interaction between the upper and lower layers of the bilayer hydrogel has a significant impact on the responsiveness of mechanical grippers [84,85,86]. Various methods are used to drive and fabricate bilayer hydrogels (Figure 4a). This demonstrates that modifying the specific sequences and crosslinking of the top and bottom hydrogel layers is a promising approach to enhancing bilayer hydrogel grippers [87].

Moreover, since bilayer hydrogel actuators composed of different materials are physically/chemically connected, the bonding strength at the interface between the two layers must meet specific requirements (Figure 4b). Therefore, the crosslinking method of the bilayer structure has become a research hotspot. One approach is physical crosslinking, where the two hydrogel layers are assembled through non-covalent forces, such as hydrogen bonding, electrostatic interactions, or hydrophobic interactions. This method does not require additional chemical reagents, is relatively mild, and has a minimal effect on the properties of the hydrogels. For instance, Fu et al. [88] fabricated nanocomposite hydrogels containing anionic poly(2-acrylamido-2-methylpropane sulfonic acid) and cationic poly(N,N-dimethylaminoethyl methacrylate methyl chloride). They then assembled hydrogels with opposite charges through electrostatic interactions to obtain a bilayer hydrogel. Ma et al. [89] developed a novel PNIPAM/TOCN/PAM thermosensitive bilayer hydrogel actuator. The two layers of the bilayer hydrogel actuator were tightly connected through hydrogen bonding between the TOCNs, allowing the actuator to maintain integrity during the bending–recovery process. In chemical crosslinking, a common method involves pre-treating elastomers with benzophenone, activating the surface, and then inducing covalent crosslinking between the pre-formed hydrogel and elastomer. Liu et al. [90] introduced double bonds to the surface of hydrogels and copolymerized them with organic gel monomers to obtain hydrogel–organogel hybrids. Although these methods show great promise, they are often limited by the types of hydrogels and involve complex preparation processes. To advance the field, Suo et al. [91] used special silane coupling agents to modify elastomers and hydrogel precursors, bonding the two layers together via siloxane bonds. In this case, a novel and versatile method was developed to create robust interfaces for hydrogel–organogel bilayer membranes. The method involved introducing nanoparticles modified with double bonds into immiscible solutions of hydrogel and organogel precursors, strategically positioned at the interface. A “one-pot” copolymerization process formed a bilayer structure in which the hydrogel–organogel phases were intricately connected through abundant covalent crosslinks [92]. Jiang et al. [93] fabricated PNIPAM-PVA@MS/PNIPAM bilayer hydrogels consisting of hydrogel microspheres (MSs) incorporated into PNIPAM-PVA composite hydrogel layers and PNIPAM hydrogel layers. Introducing PNIPAM-based MSs can enhance the structural contrast between the two layers, resulting in a faster crosslinking rate. Layer-by-layer casting involves sequentially depositing two different hydrogel solutions into a mold, allowing them to interact at the interface and form a bilayer structure. This method is relatively simple but requires precise control over the deposition time and conditions of each layer to ensure optimal interfacial bonding. Photolithography is also a popular crosslinking technique that employs patterned etching on the hydrogel surface, followed by filling the pattern with a second hydrogel to create a bilayer structure. This method allows for precise control over the shape and dimensions of the bilayer structure, making it suitable for fabricating complex hydrogel scaffolds. Additionally, template methods are commonly used, involving the use of templates with specific structures to mold one hydrogel, followed by overlaying a second hydrogel on its surface to form a bilayer structure. Templates can include porous materials or microstructure molds. For instance, using a polycarbonate membrane with a microporous structure as a template, one can fill the micropores with a hydrogel (e.g., chitosan) and then coat the membrane surface with another hydrogel (e.g., gelatin). After removing the template, a bilayer hydrogel scaffold is obtained with unique gripping properties due to its microporous structure, enabling precise manipulation of small objects. However, recent studies have shown that P(MAAm-*co*-MAAc) and PNIPAM are quite rigid. To enable the transition between rigid–flexible and rigid–stretchable states of fixtures made from these two hydrogels simply by changing temperature, a new hydrogel adhesion method needs to be developed, as existing topological adhesion methods are not suitable for such rigid hydrogels. To address this, Koo et al. [94] developed a new type of hydrogel adhesion method called split-brushing adhesion, achieving an adhesion energy of 1221.6 J/m^2^, which is 67.5 times higher than the adhesion energy obtained by other topological adhesion methods. This enabled the perfect combination of the bilayer hydrogel. In summary, research on bilayer hydrogel structures reveals that different hydrogels possess certain variations in their physical and chemical properties, making the selection of compatible hydrogel materials and appropriate composite methods crucial for the structural modification of hydrogel grippers.

Patterned structures arise from the alteration of hydrogel composition, which affects the swelling, swelling rate, and degree of swelling of the hydrogel layers, leading to differential volumetric changes between the layers and resulting in unique structures. The thickness of the layers directly influences the volumetric changes of the bilayer. Therefore, the deformation rate can be adjusted by varying the composition and thickness of the layers, while shape changes can be programmed through the design of the bilayer hydrogel’s position and shape or by constructing different hydrogel components. Compared to other structures, patterned hydrogels offer high control over induced deformations, enabling the creation of complex 3D shapes. Additionally, predefined differential swelling or responsive regions can be incorporated to achieve programmable shape changes. Nature has consistently inspired the development of hydrogel actuators. For example, an artificial “starfish”, resembling natural starfish found in the ocean, utilizes a PCPP double-hydrogel to fabricate a starfish-like actuating device. This device demonstrates exceptional performance, with its five “legs” synchronously bending under localized laser irradiation (5.0 W power, 15.0 mm spot diameter) in less than one second. Notably, it also rapidly returns to its original shape in under one second once the laser is removed (Figure 4c) [95]. Similarly, the hydrogel flower is a beautiful structure, where researchers assembled a temperature-responsive hydrogel flower using three layers of hydrogel petals of different sizes and fixed a pearl in the center. It was observed that the hydrogel flower rapidly opened and closed its petals in response to temperature changes, resembling a flower gradually blooming (Figure 4d). In a similar fashion to the artificial starfish, Wang et al. [96] created a biomimetic sea anemone actuator composed of a magnetically driven flower stem and a light-driven flower. Under the influence of an external magnetic field, the actuator could swing, mimicking the drifting motion of a sea anemone in water. A light-driven flower, made of hydrogel, could simulate the contraction and extension of a sea anemone (Figure 4e). Furthermore, straight seed pods may twist into a helical structure when dehydrated, which has encouraged exploration of hydrogel actuators with anisotropic structures in the plane to achieve complex 3D structures (Figure 4f) [97]. Photolithography is an effective method for rendering hydrogel sheets with chemically distinct regions. The resulting hydrogel sheets can undergo pre-programmed 3D shape transitions to obtain various shapes, including helical and conical forms. This demonstrates that patterned structures, utilizing photolithography, enable anisotropic hydrogel actuators with multiresponsive 3D complex shape deformations, inspiring the design and manufacturing of more intelligent hydrogel actuators.

Compared to widely fabricated bilayer structures, multilayer structures have gained extensive attention in the field of hydrogel grippers due to their superior performance in recent years. Luo et al. [98] successfully prepared a four-arm hydrogel gripper responsive to infrared light through structural design inspired by the adhesive layers of gecko toes [99]. The adjusted four-arm gripper is divided into three parts: a gripping section, a release section, and an adsorption section. The upper surface of the gripping section is made of pure PDMS, while the lower surface is composed of p(AAm-*co*-NIPAAm-*co*-TA)-Fe. The adhesive properties of the GFe/PDMS surface in the middle part improve the grip on objects (Figure 4g). The crosslinking of multilayer materials endows the four-arm gripper with strong infrared responsiveness and adhesive gripping capabilities. Recently, a three-layer hydrogel capable of reversible self-folding has been developed. A thermoresponsive hydrogel layer is sandwiched between two patterned thin rigid layers, and the stress generated during swelling drives the bending of micron-scale hinges, changing the shape of the hydrogel sample. By appropriately designing the pattern of the rigid layer based on the same crease pattern used in hand-folded origami, Hayward et al. [100] achieved a “new flapping bird” shape from a square hydrogel after swelling the sample in 22 °C water. The hydrogel sample exhibits reversible folding–unfolding shape changes when cooled and heated (Figure 4h). The design of multilayer structures, to some extent, overcomes the limitations of bilayer structures in terms of mechanical properties and functionality. However, the crosslinking of multilayer hydrogels imposes higher requirements on material selection.

**Figure 4 biomimetics-09-00585-f004:**
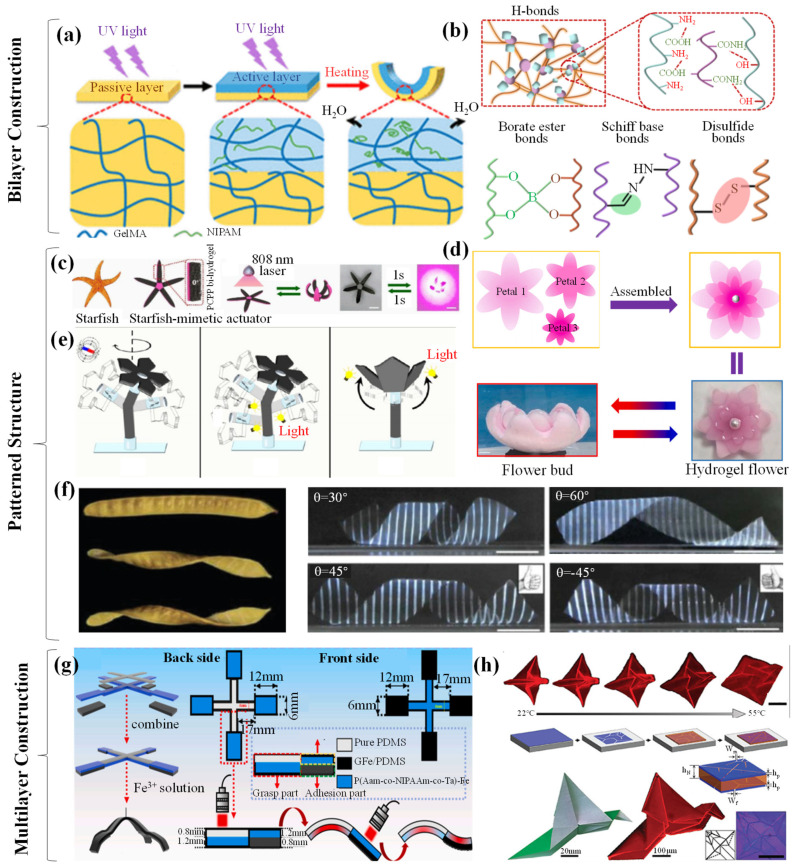
(**a**) Illustration of the bilayer hydrogel [101]. Copyright 2024, John Wiley and Sons Inc. (**b**) Preparation of hydrogels using physical and chemical crosslinking methods [102]. Copyright 2024 Elsevier. (**c**) Anisotropic structure of the starfish-mimicking actuator device and its performance [101]. Copyright 2024 John Wiley and Sons Inc. (**d**) Schematic diagram of the hydrogel petal layers [93]. Copyright 2022 Elsevier. (**e**) Structural diagram and actuation mechanism of the bionic sea anemone actuator [96]. Copyright 2023 American Chemical Society. (**f**) Stages of seed pod twisting and the spiral images generated by anisotropic hydrogel sheets. Copyright 2019 Wiley-VCH Verlag. (**g**) Schematic diagram of the adjusted four-arm gripper structure [103]. Copyright 2024, Science China Press. (**h**) Three-layer hydrogel [104]. Copyright 2018 Wiley.

Most hydrogels are synthesized through the polymerization or assembly of molecular components uniformly dissolved in an aqueous medium, resulting in structures that are generally isotropic [105]. However, the directional structure of hydrogels refers to a structure formed by the orderly arrangement of components inside the hydrogel in a specific direction. The formation methods of such directional structures have been a topic of ongoing research. Hydrogels can form directional structures induced by shear forces. Recently, Lewis et al. [106] created a programmable biomimetic hydrogel actuator. They used an ink containing acrylamide monomer, cellulose, and other components. During the printing process, when the ink flows through the deposition nozzle, the shear forces of the fluid induce the hydrogel to form a directional structure. Through this process, the fibrils undergo shear-induced alignment, making the printed filaments anisotropic in terms of stiffness and expansion behavior along the longitudinal direction (along the printing path). The flowers printed using this biomimetic 4D printing technology, which mimic plant structures, can change shape in water and exhibit complex 3D forms (Figure 5a). Additionally, external forces can be used to induce the formation of directional structures. Goudu et al. [48] used hydrogel precursors composed of light-crosslinked collagen derived from porcine extracellular matrix and superparamagnetic iron oxide nanoparticles (SPIONs). Using polydimethylsiloxane (PDMS) as a mold for grippers, they solidified the hydrogel precursors added into the PDMS mold through photopolymerization to make milli-grippers. By applying a magnetic field generated by an external permanent magnet, the SPION chains inside the hydrogel are directionally assembled, forming a directional structure (Figure 5b). Moreover, the research by these scientists demonstrates several advantages of directional structures. Firstly, external forces (e.g., magnetic forces) or shear forces can precisely control the alignment of nanofillers. Secondly, the directional alignment of nanofillers can induce anisotropic expansion behavior. Directional structures can also endow hydrogels with various performance enhancements, such as anisotropic mechanical strength (e.g., stiffness in different directions) and responsiveness to external stimuli (e.g., magnetic fields). These capabilities allow hydrogels to achieve customized shape transformations and actuation, enabling the design of complex biomimetic structures. However, the fabrication of directional structures also faces challenges, such as whether the processing technology and materials can meet the requirements. Additionally, the thickness of the hydrogel affects whether the nanofillers can align uniformly [101].

In addition to the aforementioned structural modifications, gradient structures represent an advanced design concept for preparing hydrogel actuators. By introducing gradients of composition, concentration, or physical properties in different regions of the material, non-uniform stress distribution is achieved within the material. This stress distribution can be converted into precise mechanical motions, such as bending, twisting, or stretching, under light stimulation [107,108,109]. Due to the ability of gradient structures to achieve complex deformation through variations in polymer chain distribution or filler concentration, and to generate differentiated responses in different areas, researchers can more precisely control the motion paths and deformation behavior of hydrogel grippers, allowing them to adapt to various complex operational environments. For example, in small medical operations or dexterous gripping tasks, gradient structures enable hydrogel actuators to perform tasks in a flexible and efficient manner, something that is difficult for traditional rigid robots to achieve [110]. The main types of gradient structure modification include thickness gradient, composition gradient, and local gradient. Among these, thickness gradient hydrogels are created by introducing a gradient in the thickness direction of the material, causing shape changes when the hydrogel swells or deswells (Figure 5c). This structure is typically formed by guiding the distribution of polymers or crosslinking agents through molds or external fields (e.g., magnetic or electric fields). Composition gradient hydrogels are created by introducing different chemical components in different regions of the material, enabling the hydrogel to exhibit different shape changes when subjected to uniform stimuli. Local gradient hydrogels are achieved by introducing different responsive characteristics in specific regions of the material, allowing localized deformation [111,112]. Studies of these three types of gradient structures have shown that through complex molecular design and precise arrangement of nanoparticles, efficient and complex shape deformation can be achieved, providing key technical support for the development of hydrogel actuators and offering potential for achieving complex shape transformations.

## 4. Stimuli-Responsive Hydrogel Grippers

Hydrogel grippers achieve grab–release of objects through responses to various external stimuli, and better understanding of stimuli-responsive mechanisms plays an essential role in fabricating excellent-performance hydrogel grippers. In this section, various external stimuli-responsive mechanisms are introduced, including thermal, electronic, magnetic, pH, ion content, electrochemical, DAN, ultrasound, multiresponsive, etc. In addition, various typical studies about stimuli-responsive hydrogel grippers’ grab and release of objects are summarized for readers to have an intuitive feeling of hydrogel grippers’ applications.

### 4.1. Thermal-Responsive Hydrogel Grippers

#### 4.1.1. Thermal-Responsive Driving Mechanisms

In the current research of hydrogel grippers, the temperature sensitivity, strong adaptability, and high controllability of thermally responsive hydrogel grippers have made them an extremely attractive and sought-after responsive approach, triggering in-depth research and exploration among scholars. This section reviews the thermally responsive methods of hydrogel grippers in three directions: high load capacity, rapid response, and photothermal response [113].

Hydrogel grippers have attracted the attention of numerous scholars with their unique properties. Among them, their high load-bearing capacity has become the focus of research. Li et al. [103] developed a bilayer hydrogel gripper that can self-lock and lift objects up to 500 times its own weight by relying on the structural grasping force. The PNIPAM hydrogel layer incorporating microgels functions as the driving layer, while the poly(acrylic acid)-calcium acetate (PAA-Ca(CH_3_COO)_2_) hydrogel layer acts as the thermosetting layer. The gripper accomplishes the grasping action via the contraction of the PNIPAM composite layer. Due to the increase in the stiffness of the PAA-Ca (CH_3_COO)_2_ hydrogel layer with increasing temperature, further increasing the temperature can lock the deformed shape of the bilayer hydrogel by the thermosetting layer to achieve grasping (Figure 6a). Under the influence of the thermosetting layer, the hydrogel gripper can lock its grasping action, thereby enhancing its mechanical properties. By increasing the modulus of the thermosetting layer, the hydrogel gripper can endure greater forces and consequently be capable of grasping heavier objects. For instance, the bilayer hydrogel actuator comprising short M5PNIPAm hydrogel strips and long PAA-Ca(CH_3_COO)_2_ hydrogel strips can deform into a hook in warm water at 50 °C. A hook weighing 0.2 g can lift a lightweight object weighing 3 g. However, if the object weighs up to 4 g, it cannot be lifted due to insufficient mechanical strength. When heated to 70 °C, the PAA-Ca(CH_3_COO)_2_ hydrogel layer undergoes hardening. Consequently, the hook is reinforced and becomes capable of lifting a 100 g heavy object, exhibiting a lifting capacity of more than 33 times. The hydrogel sheet can deform and wrap around a 48 g screw in water at 50 °C and can be lifted when the temperature rises (Figure 6b). From 50 to 70 °C, the modulus of the PAA-Ca(CH_3_COO)_2_ layer increases by approximately 6 times, and the ratio of the hydrogel hook to the object mass increases from 15 to 500 (Figure 6c), indicating that a slight change in the modulus of the thermosetting layer can significantly change the mechanical strength of the hydrogel actuator. This strategy promotes the development of advanced soft actuators and robotic technologies. Koo et al. [94], inspired by the human hand, developed an all-hydrogel gripper that can withstand a weight of more than 47.6 times its own weight by relying on adhesion [114]. The researchers employed the method of monomer diffusion and polymerization to attain adhesion between hydrogels. By applying an initiator on one hydrogel surface and a catalyst on the other hydrogel surface, a polymer network can be formed between the two hydrogel surfaces, thereby realizing adhesion. In comparison with the adhesion method based on polymer diffusion, higher adhesion energy can be obtained through monomer diffusion and polymerization. The gripper is fabricated from P(MAAm-*co*-MAAc) and PNIPAM. When the temperature rises, the cartilage hydrogel and the flexor tendon hydrogel undergo volume contraction, whereas the bone hydrogel remains relatively stable. This leads to the gripper presenting a curved state. When the temperature decreases, the cartilage hydrogel and the flexor tendon hydrogel expand, thereby restoring the gripper to the extended state. The finger is in a softly curved condition. In the single-step cooling path at 45 °C, the temperature of the finger drops to 4 °C within 30 s. Owing to the insufficient expansion time of the flexor tendon hydrogel, the finger switches from a soft bend to a hard bend, while the bending angle remains unchanged. The resulting curved finger possesses a high load-bearing capacity. In the two-step cooling path, the temperature of the all-hydrogel finger drops to 32 °C and is maintained for 30 h, enabling the flexor tendon hydrogel to fully expand. At this temperature, the cartilage hydrogel is soft, permitting the finger to stretch. Even under the bending, the all-hydrogel finger can withstand a load of 46.7 times its own weight (Figure 6d). Subsequently, the researchers assembled five fingers and a palm to demonstrate that the all-hydrogel fixture possesses a high load-bearing capacity. The rigid bending state of the all-hydrogel hand is obtained via the single-step cooling path, while the rigid stretching state is achieved through the two-step cooling path. Additionally, the researchers illustrated that this hand overcomes the limitations of existing hydrogel grippers in the stiff bending state and is capable of lifting heavy objects. This hand can lift a heavy object weighing 47.6 times its own weight. When the adhesion force between the fingertip of the hand and the object is strong enough, this hand can grasp a 100 g object (Figure 6e). With the continuous advancement of technology, high load-bearing capacity hydrogel grippers may open up entirely new application scenarios and become a powerful driving force for the development of related industries.

After understanding the outstanding performance of high load capacity, focus will be turned to another important characteristic of hydrogel grippers: rapid response. Spratte et al. [115] optimized the driving function by means of interconnected microchannels and introduced a random network of interconnected microchannels into the 3D structure of a PNIPAM hydrogel gripper that relies on structural grasping force. They infiltrated the PNIPAM-based hydrogel precursor solution into a highly porous sintered 3D scaffold composed of micron-sized zinc oxide tetrapods, which are referred to as templates. After the polymerization was completed, zinc oxide was dissolved chemically, thus generating a replica of the template in the hydrogel, namely, a hydrogel with interconnected hollow microchannels (Figure 6f). Through such microstructure improvement, the researchers confirmed that the thermal response contraction of the material proceeds at an extremely high rate. Compared with the traditional bulk PNIPAM, this design significantly improves the performance of the gripper and lays the foundation for new applications in thermal response contraction systems and devices. Li et al. [116] successfully prepared an ultrafast thermally responsive gripper (responding within only 9 s in water at 60 °C) that relies on structural grasping force and a high-strength ultrafast thermally responsive VSNPs-P(NIPAM-*co*-AA) hydrogel containing multivalent vinyl-functionalized silica nanoparticles (VSNPs). The grafted polymer chains tend to curl and form hydrophobic aggregations around the relatively fixed multivalent VSNP crosslinking points, thereby generating some water transport channels within the hydrogel, which is conducive to drainage (Figure 6g). The hydrogel contracts gradually from the surface inward to achieve rapid response. The content of VSNPs has a significant impact on the contraction behavior of the thermally responsive PNIPAM-based hydrogel. The contraction curves of the length change of PNIPAM hydrogel membranes with different VSNP contents over time were shown. The PNIPAM hydrogel crosslinked by 0.5 wt% MBAA shows a slow response rate at the initial stage. The data present the changes in the relevant volume contraction rate of PNIPAM hydrogels with different VSNP contents over time. In contrast, the VSNPs-PNIPAM hydrogel containing 0.25–0.5 wt% VSNPs and only 0.01 wt% MBAA shows a significant increase in the initial response rate (Figure 6h). Overall, for PNIPAM hydrogels with free-hanging polymer chains, they can contract more rapidly because their conformations are more prone to change due to the more flexible and movable polymer chains. Hydrogel grippers with rapid response characteristics can be applied in more practical scenarios, such as intelligent flexible actuators or soft robots.

**Figure 6 biomimetics-09-00585-f006:**
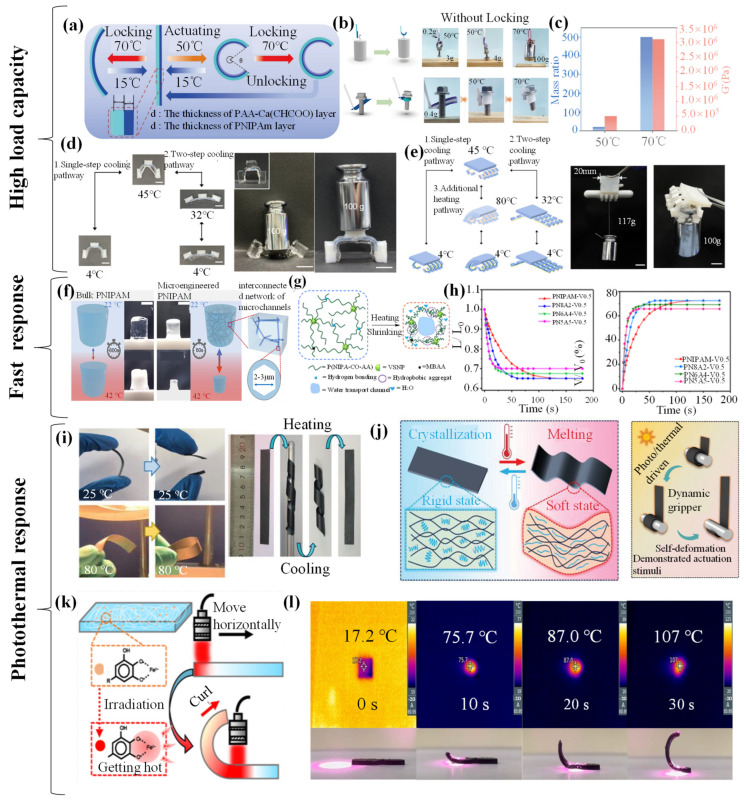
(**a**) Schematic illustration presents the shape transformation behavior of an M5PNIPAm/PAA-Ca(CH_3_COO)_2_ bilayer hydrogel strip. The bilayer hydrogel strip is capable of deforming at 50 °C, and its deformed shape can be locked at 70 °C. If the bilayer hydrogel strip is locked directly at 70 °C, the extent of deformation can be significantly reduced. (**b**) Schematic of a bilayer hydrogel grippers’ grab and release of objects in response to temperature. (**c**) Comparison is made regarding the modulus and lifting force of the M5PNIPAm/PAA-Ca(CH_3_COO)_2_ bilayer hydrogel at 50 and 70 °C [103]. Copyright 2024, Science China Press. (**d**) A working cycle of the all-hydrogel finger: single- and two-step cooling pathways. Photos of load bearing with a soft hydrogel finger made of the PAAm hydrogel (top) and our all-hydrogel finger (bottom). All scale bars are 10 mm. (**e**) Schematic illustration of three pathways of the all-hydrogel hand for three distinct states of bending and stiffness: the single- and two-step cooling pathways and the additional heating pathway. Photo of the all-hydrogel hand to lift a heavy object with high stiffness indirectly and photo of the all-hydrogel hand to lift a heavy object with high stiffness directly. All scale bars are 10 mm [94]. Copyright 2023, the Royal Society of Chemistry. (**f**) Sketches and photographs of bulk pNIPAM (on the left) and microengineered, channel-containing pNIPAM (on the right). Compared with the bulk material, the microengineered material undergoes a much larger volume change during the phase transition at the LCST and also reacts significantly faster [116]. Copyright 2021, Wiley-VCH GmbH. (**g**) Illustration of the network structure transformation when heating the thermoresponsive PNIPAM-based hydrogels composed of VSNPs, showing the facilitated hydrophobic aggregation of the mobile grafted polymer chains. This can create water transport channels inside the hydrogels and help in the rapid expulsion of water. (**h**) Shrinking curves for the length change (L/L0) versus time, and volume shrinking ratio (ΔV/V_0_) versus time of the PNIPAM hydrogels with different VSNP contents in 60 °C water [116]. Copyright 2022, American Chemical Society. (**i**) Photographs demonstrating the shape memory behavior of the PEG/ANF/GNP film actuated by heat at 25 and 80 °C. (**j**) Schematic diagram of the microstructure change showing the internal PEG polymer chains in the crystallization and melting state [117]. Copyright 2024, Wiley-VCH GmbH. (**k**) Schematic diagram of the infrared driving mechanism of p(AAm-*co*-NIPAAm-*co*-TA)-Fe. (**l**) Pictures depicting the change in the surface temperature of 5 wt% NIPAAm p(AAm-*co*-NIPAAm-*co*-TA)-Fe with infrared irradiation time. And NIR driver diagram of p(AAm-*co*-NIPAAm-*co*-TA)-Fe [98]. Copyright 2022, Elsevier Ltd.

In addition to the discussion on the individual thermal response of hydrogel grippers, there are also studies on photothermal coupling. The mechanism by which light energy is converted into thermal energy to trigger changes in the hydrogel structure, as well as its advantages such as precise controllability and remote operation, has opened up a broader research field for the application of hydrogels. Han et al. [117] successfully prepared a phase-change layered film by in situ filling with polyethylene glycol (PEG) in an aramid nanofiber (ANF)/graphene nanoplatelet (GNP) network. The excellent light absorption and photothermal effect of graphene endow the phase-change material film with outstanding photothermal capacity. The dynamic mechanical properties of the PEG/ANF/GNP film determined by the dynamic thermomechanical analyzer (DMA) test reveal its temperature-dependent storage modulus, causing it to change from brittleness to toughness with the change in the ambient temperature (Figure 6i). Moreover, the phase-change layered film exhibits temperature-dependent shape memory behavior based on temperature-dependent mechanical properties. The PEG/ANF/GNP film deforms when heated, takes shape when cooled, and restores its shape when heated again. This behavior is mainly due to the reversible crystallization and melting transitions of PEG within the phase-change layered film. When the film is cooled, the ordered crystal structure limits the movement of the chains and the ANF/GNP network, making the phase-change layered film have a rigid state. Conversely, when heated, the molten PEG chains eliminate these movement restrictions, thereby imparting flexibility to the layered film. Given this temperature-dependent shape memory behavior and the photothermal conversion characteristics of the PEG/ANF/GNP layered film, a light/thermal-driven dynamic gripper relying on structural grasping force can be designed for grasping and moving objects at low temperatures and releasing them at high temperatures (Figure 6j). This phase-change film has great potential in practical applications such as thermal management and dynamic hydrogel grippers. Luo et al. [98], inspired by the microstructure of gecko toes, designed and fabricated a bilayer-structured adhesive-driven biomimetic gecko toe adhesion surface hydrogel gripper. The researchers conducted photothermal tests on the hydrogel, and the infrared driving mechanism is shown Figure 6k. Under the irradiation of 4 W/cm^2^ NIR (808 nm), the surface temperature of p(AAm-*co*-NIPAAm-*co*-TA)-Fe rises, promoting the effective heating of the hydrogel and the removal of moisture from the matrix to achieve the corresponding action. The introduction of tannic acid (TA) and the immersion of trivalent iron ions into p(AAm-*co*-NIPAAm-*co*-TA)-Fe aim to enhance the photothermal effect of the p(AAm-*co*-NIPAAm-*co*-TA)-Fe material and achieve chelate with Fe^3+^. When the p(AAm-*co*-NIPAAm-*co*-TA)-Fe material is exposed to NIR light, TA absorbs NIR light and generates thermal energy, causing the temperature of the material to rise. At the same time, by immersing the Fe^3+^ ions in the p(AAm-*co*-NIPAAm-*co*-TA)-Fe material, the light absorption capacity of the material can be increased, further improving the temperature rise rate and driving effect of the material. Fe^3+^ ions can form complexes with TA to enhance the photothermal conversion efficiency of TA. With the aid of a thermal imager, the change in the surface temperature of 5 wt% NIPAAm p(AAm-*co*-NIPAAm-*co*-TA)-Fe over time can be seen. The surface temperature of 5 wt% p(AAm-*co*-NIPAAm-*co*-TA)-Fe can increase from 17.2 to 107 °C within 30 s, while p(AAm-*co*-NIPAAm-*co*-TA) is not affected by the same irradiation. The NIR driving demonstration of 5wt % NIPAAm p(AAm-*co*-NIPAAm-*co*-TA)-Fe was shown. It can be seen that p(AAm-*co*-NIPAAm-*co*-TA)-Fe has excellent photothermal performance, which forms the basis for preparing the bilayer biomimetic driving surface of the flexible grippers (Figure 6l). The bilayer biomimetic surface proposed in this work has broad application prospects in the field of intelligent adhesion systems and flexible arm grippers and has potential applications in the field of infrared-responsive hydrogel grippers.

To sum up, in the current numerous research works on thermally responsive hydrogels, PNIPAM is mostly used as the base material in most cases. The fundamental reason for this widespread situation lies in the unique and crucial properties of PNIPAM, namely its temperature sensitivity and good chemical stability. Therefore, thermally responsive hydrogel grippers have remarkable performance, such as precise temperature sensitivity, outstanding deformation ability, etc. With its outstanding performance and innovative design, the thermally responsive hydrogel gripper has undoubtedly become a shining star in the field of hydrogel grippers. With the continuous deepening of research and the continuous progress of technology, it will bring more outstanding application achievements to more fields in the future.

#### 4.1.2. Thermal-Responsive Grab and Release of Objects

In the continuous evolution and development of hydrogel grippers, thermally responsive hydrogel grippers have emerged prominently in various application fields due to their unique performance characteristics. The thermal responsiveness endows these grippers with a keen perception of environmental temperature and a precise response capability, thereby providing innovative and efficient solutions for addressing a range of practical problems.

This section first focuses on the task of gripping fragile and deformable objects in the air. Li et al. [103] pointed out that achieving robust grasping remains a significant challenge. To address this, they developed a self-locking bilayer hydrogel gripper to achieve effective and stable grasping of heavy objects. The researchers designed a six-arm hydrogel gripper, which includes a PNIPAM hydrogel layer containing microgels as the driving layer and a PAA-Ca(CH_3_COO)_2_ hydrogel layer as the thermally hardening layer. This gripper can deform and grasp an egg yolk in 50 °C warm water (Figure 7a), and the grasping action can be locked by enhancing the hardness of the PAA-Ca(CH_3_COO)_2_ layer. Thanks to the softness of the M5PNIPAm layer, the hydrogel gripper can successfully lift the heavy egg yolk without crushing it. This design approach is expected to open a new avenue for improving the gripping force of soft actuators. Similarly, Li et al. [116] prepared an ultrafast thermally responsive VSNPs-P(NIPAM-*co*-AA) hydrogel. The integrated gradient hydrogel, obtained by self-healing of the thermally responsive VSNPs-PN6A4 hydrogel and the high-strength VSNPs-PAA-Fe^3+^ hydrogel, is applied as a thermally responsive soft actuator (e.g., for grasping a toy). Before testing, it was cut into a cross-shaped gripper. When transferred from 4 to 60 °C water, the gripper quickly bent and curled, completing the wrapping and capturing motion within 9 s. After 9 s in 60 °C water, the tensile strength of the relevant hydrogel was enhanced, allowing the gripper to grasp heavier objects (Figure 7b). The integrated gradient hydrogel actuator exhibits rapid thermal response, outperforming most existing hydrogel actuators. In comparison, relying on the self-healing of the two hydrogels to create the integrated gradient hydrogel actuator is simple and time-saving, and due to its inherent properties, it is more promising as a practical smart flexible actuator or soft robot.

Hydrogel grippers, as an innovative tool, have demonstrated unique advantages in the field of object grasping and manipulation. Their performance in gripping objects of varying weights in water is particularly noteworthy. Ji et al. [47] aim to address the problem of simplifying the preparation of structures by precisely controlling shape deformation. They employed 3D printing technology to create a thermally responsive hydrogel gripper with complex and controllable shape deformation. The researchers printed a thermally responsive gripper and, upon immersing it in room-temperature water, the branches curled inward due to self-bending. During the bending process, a hollow sphere weighing approximately 0.15 g was easily captured by the branches, with a capture time of about 7 s. In warm water, however, the branches of the gripper began to slowly return to their original shape, resulting in the release of the hollow sphere in about 70 s (Figure 7c). Although the release time in warm water was much longer than the capture time, it was still shorter than the release time in air. This research is expected to pave the way for new methods in the production of hydrogel grippers. Liu et al. [118] aimed to develop a bioinspired gradient hydrogel with self-sensing and actuation capabilities. They successfully created a novel multifunctional self-sensing actuated gradient hydrogel that exhibits ultrafast thermal response and enhanced photothermal efficiency. This PSM hydrogel was fabricated through in situ copolymerization of NIPAM and sodium alginate (SA) monomers in a molybdenum dioxide (MoO_2_) dispersion. Utilizing the ultrafast thermal response driving characteristics of the PSM hydrogel, the researchers successfully manufactured a biomimetic four-arm gripper (similar to a “Gold Miner”) designed to grasp target objects (Figure 7d). When the gripper was placed in water at 50 °C, it rapidly bent downward and successfully grasped a metal sheet within 15 s. The novel and efficient high-performance multifunctional self-sensing actuated gradient hydrogel preparation method proposed in this study paves the way for the development of the next generation of intelligent interactive haptic soft materials.

In recent years, the emergence of photothermally responsive hydrogel grippers has garnered significant attention due to their unique properties, offering a novel solution for object gripping. Luo et al. [98], inspired by the microscopic structure of gecko toes, designed and fabricated a dual-layer structure biomimetic gecko toe adhesive surface. Based on this, they optimized the dual-layer structure to develop a four-arm gripper that could grasp/release objects with unilateral irradiation. The researchers continuously improved the four-arm gripper and explored using near-infrared light to remotely control the adjusted four-arm gripper. After about 120 s of infrared radiation, the gripping part of the adjusted four-arm gripper contracted and, relying on van der Waals forces, grasped the object, while the shear and adhesive forces of the adhesive part prevented the object from falling. When the infrared radiation continued to irradiate the releasing part of the adjusted four-arm gripper for about 60 s, the p(AAm-*co*-NIPAAm-*co*-TA)-Fe would dehydrate and bend due to the heat, achieving object release (Figure 7e). The dual-layer biomimetic surface proposed in this study has broad application prospects in the fields of intelligent adhesion systems and flexible arm grippers. Similarly, Zhang et al. [119], inspired by the rolling behavior of gecko feet, integrated a thermally responsive hydrogel layer, a mushroom structure array inspired by gecko feet, and a mussel-inspired copolymer adhesive coating to successfully develop a novel multilayer self-peeling switchable dry/wet adhesive (SPSA). Subsequently, the gecko foot-like behavior of the SPSA device was demonstrated for transferring objects from a cold water bath to a hot water bath (Figure 7f). To reduce interference from the interfacial capillary action between the object and water, a small SPSA device with short bristles was used for demonstration. The SPSA device successfully transferred triangular PMMA plastic with a 12 g load from a 20 °C cold water bath to a 50 °C hot water bath. This design is expected to have broader applications in engineering fields and provides a new avenue for developing intelligent adhesive hydrogel grippers.

In addition to the above research, thermomagnetic response has also emerged as a new research direction. Kobayashi et al. [120] developed a thermomagnetically responsive multifunctional soft robotic gripper with biocompatibility and biodegradability. The gripper’s magnetic properties endow it with the ability to perform the crucial robotic task of picking and placing. This task is common in macrorobotics, for instance, on assembly lines. It is also of great importance in medical fields, such as during stent placement or the delivery of embolization devices. Sequential optical images prominently showcased the P(OEGMA-DSDMA)/Fe_2_O_3_ embedded P(AAm-BAC) gripper’s ability to pick up and place a soft object (soft tofu, estimated modulus in the range of 1–5 kPa, similar to the modulus of many endothelial and matrix tissues, as well as gels) (Figure 7g). This method holds significant potential for applications in deformable biomedical devices and surgical robotic applications.

In summary, the application of thermally responsive hydrogel grippers has demonstrated broad prospects and tremendous potential. As research deepens and technology advances, their application in more critical fields will continue to expand and deepen. This will play an increasingly important role in driving technological progress and addressing practical needs, leading hydrogel grippers toward a more impactful future.

### 4.2. Electronic-/Magnetic-Responsive Hydrogel Grippers

#### 4.2.1. Electronic-/Magnetic-Responsive Driving Mechanisms

Hydrogels, as crosslinked polymer networks capable of absorbing and retaining significant amounts of water without dissolving, exhibit remarkable swelling and deswelling properties. These characteristics enable considerable shape changes and mechanical responses under external stimuli such as electric fields, magnetic fields, or electromagnetic coupling. The volume changes driven by physical or chemical transformations represent a core mechanism for the actuating functions of hydrogels. This section delves into the latest advancements in the domains of electric, magnetic, and electromagnetic driving, showcasing the technological allure and extensive application prospects [121,122].

In the expansive field of smart materials research, electrically responsive hydrogels stand out for their unique electro-driven characteristics, presenting significant applications in the realm of hydrogel grippers. These hydrogels exhibit expansion or contraction behavior in response to solvent induction or externally applied electric fields. The grasping mechanism of electric response hydrogel grippers primarily involves ion movement within the electric field, along with the arrangement and distribution of ions in the hydrogel swelling medium. To optimize the electro-driven performance of hydrogel grippers, the properties of the hydrogel and the materials and relative positioning of the electrodes are crucial [123]. Shin et al. [124] fabricated electrically responsive poly(4-vinylsulfonic acid sodium/2-hydroxyethyl methacrylate/acrylamide) (P(VBS/HEMA/AAm)) and poly(4-vinylsulfonic acid sodium/2-hydroxyethyl methacrylate/acrylic acid) (P(VBS/HEMA/AAc)) hydrogels by finely tuning the monomer ratio and crosslinking density, achieving systematic optimization of hydrogel driving performance in an electric field. Notably, these hydrogels can achieve rapid bending deformation at voltages as low as 10 V, with driving performance adjustable through various factors (Figure 8a). These hydrogels not only possess a high gel fraction (>65%) and substantial equilibrium water content (EWC > 90%) but also demonstrate sensitivity to parameters such as crosslinker content and monomer combinations. Experimental data reveal that the maximum bending angle of the hydrogel depends on the PEGDA content (Figure 8b). Importantly, these hydrogels exhibit exceptional performance under low electric fields, responding rapidly (bending over 100° within one minute) while maintaining good biocompatibility (cell viability > 84%), indicating promising prospects for their application in soft actuators. Nonetheless, some limitations remain. The voltage dependence of hydrogels and their response to multiple environmental factors, such as ion concentration, electrode spacing, and electric field strength, collectively restrict their application potential in low-power, miniaturized devices and variable environments. Further research is necessary to improve these aspects.

Building upon the successful optimization of the material properties of electroresponsive hydrogels, Ko et al. [125] have achieved a significant leap in the performance of hydrogel actuators through innovative electrode design and actuation mechanisms. They developed a high-performance hydrogel actuator. They utilized a unique layer-by-layer assembly technique to construct interconnected, crack-resistant, and mechanically flexible metallic electrodes on the hydrogel’s surface. This actuator cleverly harnesses the electrokinetic pumping effect induced by an electric field, successfully generating powerful hydraulic flow within the hydrogel, allowing for rapid expansion. During this process, the actuator achieved up to 20% actuation strain and an energy density of 1.06 × 105 J m^−3^. The mechanism is based on the application of approximately 3 V to the crack electrodes, which promptly activates electrokinetic pumping, resulting in significant hydraulic fluid flow inside the hydrogel. This fluid flow is not only reversible but also highly potent. It exerts a direct influence on the hydrogel and gives rise to substantial expansion deformation (Figure 8c). The electrokinetically operated (EOP) hydrogel actuator displays exceptional performance in driving effects. It operates at low voltage, responds quickly, and features a holistic structural-based motion mechanism that facilitates complex and adaptable multi-degree-of-freedom movements. These characteristics offer the EOP gel actuator extensive application potential in fields such as soft robotics and wearable devices, providing efficient and flexible driving solutions through reversible changes in the overall fluid state under hydration. Additionally, research has explored the application potential of stimulus-responsive polyelectrolyte hydrogels in soft robotics, which expand and deform through electric field modulation of ion distribution, proposing new ideas for the design of walking gel actuators [126].

The application domains of hydrogel actuators have expanded from electric field driving to magnetic field manipulation. Magnetically responsive hydrogel grippers primarily achieve functionality by incorporating ferromagnetic or paramagnetic functional particles into a polymer matrix. Depending on their magnetization characteristics, the added particles fall into two categories, soft magnetic additive particles and hard magnetic additive particles. Soft magnetic particles, such as iron, alloys, and oxides, exhibit high magnetic permeability and saturation magnetization [127]. In contrast, hard magnetic particles typically demonstrate significant magnetic hysteresis and are more suitable for permanent magnets, such as neodymium–iron–boron (NdFeB) magnets [128]. The introduction of magnetic particles generally does not require additional crosslinking agents. These particles often participate in covalent or synergistic crosslinking with the polymer during the crosslinking process. For instance, Tian et al. [129] developed a distinctive magnetic shape memory hydrogel via a one-pot polymerization method, cleverly combining MAAm, MAAc, PVA, MPS-modified Fe_3_O_4_ nanoparticles, and potassium persulfate (KPS) as an initiator to prepare a homogeneous solution. Subsequently, upon heating at 60 °C for 8 h, polymerization occurred within the reaction vessel, yielding a magnetic hydrogel embedded with Fe_3_O_4_ nanoparticles. The mechanism of this work involves magnetic hydrogels maintaining rigidity at room temperature due to the presence of Fe_3_O_4_ particles, which resist deformation. When exposed to an alternating magnetic field (AMF), the Fe_3_O_4_ generates heat, causing the hydrogel to exceed the glass transition temperature (Tg). This effect softens the material, enabling it to deform in response to the driving magnetic field. Once the AMF is removed, the hydrogel cools below Tg, increasing its modulus and retaining the deformed state without the need for continuous magnetic drive (Figure 8d). This hydrogel demonstrates excellent mechanical properties and precisely controlled stiffness through non-covalent hydrogen bonding between PVA and P(MAAm-*co*-MAAc). The incorporation of Fe_3_O_4_ particles endows the material with the ability to deform flexibly under an alternating magnetic field, followed by automatic hardening to maintain the desired shape upon removal of the magnetic field. This characteristic enables accurate shape retention and object grasping without the need for a continuous magnetic field. However, it also faces challenges such as uncertain long-term stability, increased complexity in preparation, and unclear shape recovery rates. Its performance can be further optimized to suit a wider range of application scenarios. Additionally, by modifying the formulation and preparation parameters, performance can be further optimized to meet diverse application scenarios. Furthermore, in the realm of magnetically driven technologies, the strategy of integrating hard magnetic soft robots with self-sensing capabilities warrants attention. This approach employs a bilayer structure of magnetic hydrogel and ion-conducting hydrogel to achieve both actuation and self-perception functions, thereby offering novel solutions for intelligent grasping and dynamic monitoring [130].

The magnetically responsive hydrogel gripper is programmable and remotely controllable, in addition to the speed, precision, and large deformations demonstrated in the work described above. Goudu et al. [48] successfully developed an innovative, fully biodegradable, untethered soft microrobotic device called the hydrogel milli-gripper. This device demonstrates the capacity to encode three-dimensional magnetic anisotropy, thus enabling static or dynamic shape programming. The milli-gripper comprises a collagen-based hydrogel network derived from pig extracellular matrix and embedded superparamagnetic iron oxide nanoparticles (SPIONs). By utilizing an external permanent magnet to directionally self-assemble SPION chains, it creates a three-dimensional magnetized structure within the hydrogel, granting the robot flexible shape-change abilities. They selected gelatin methacrylate hydrogel as the primary material, capitalizing on its biodegradability and mechanical strength. Through the combined effects of a photoinitiator and ultraviolet light, they constructed a network with a complex 3D porous structure (Figure 8e). Subsequently, SPIONs were uniformly dispersed and firmly anchored within a gel matrix, resulting in a three-dimensional magnetic anisotropic nanocomposite hydrogel milli-gripper. Notably, this hydrogel milli-gripper displayed excellent biocompatibility and biodegradability in physiological environments. Upon exposure to matrix metalloproteinase-2 (MMP-2), it can completely degrade, and the degradation products are safe and non-toxic to biological systems, which provides significant assurance for biomedical applications. However, this study also has certain limitations. The magnetic field dependency restricts autonomy and flexibility. The manufacturing process demands precise operations and robust magnetic field control. Furthermore, the current functionality remains somewhat singular and requires further optimization.

Additionally, in response to the demand for miniaturization of robotic grippers, Shao et al. [131] successfully developed a magnetic-driven three-finger microgripper using 3D printing technology. This gripper utilizes magnetic actuation for seamless operation in complex environments and optimizes the balance between mechanical compliance and magnetic driving force through microcontinuous liquid interface production technology (μCLIP). This achievement highlights significant potential in micromanufacturing, assembly, and biological and biomedical applications, such as the precise manipulation of living cells and soft tissues.

**Figure 8 biomimetics-09-00585-f008:**
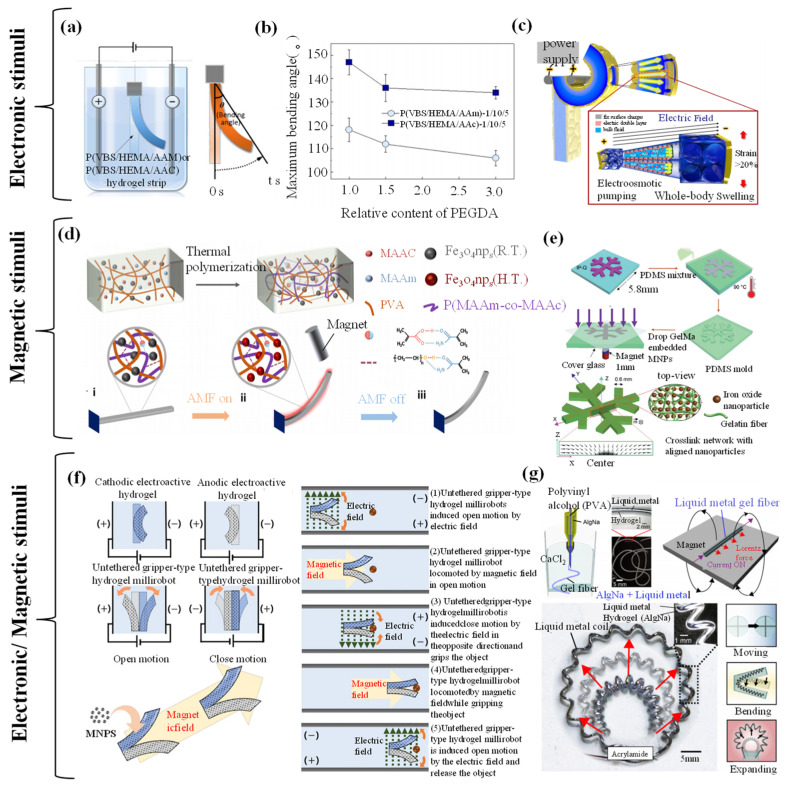
(**a**) Schematic diagram of hydrogel bending actuation under the influence of an electric field. (**b**) Maximum bending angle of P(VBS/HEMA/AAm)-1/10/5 and P (VBS/HEMA/AAc)-1/10/5 hydrogels, depending on PEGDA content [124]. Copyright 2021, American Chemical Society. (**c**) Schematic of the working mechanism of the EOP hydrogel driver using LbL to assemble a crack-inducing electrode [125]. Copyright 2020, American Chemical Society. (**d**) Schematic synthesis of magnetic hydrogel and its working mechanism [129]. Copyright 2024, the Royal Society of Chemistry. (**e**) Three-dimensional directional self-assembly of SPION is achieved by a non-uniform magnetic field generated by an external permanent magnet to form a soft internal structure with specific magnetic anisotropy [48]. Copyright 2020, the authors. (**f**) Robots with built-in magnetic nanoparticles work under electric and magnetic fields [132]. Copyright 2020, IOP Publishing Ltd. (**g**) Robots with built-in magnetic nanoparticles work under electric and magnetic fields [133]. Copyright 2022, the authors.

In addition to the investigation of the separate electro- and magnetic responses of hydrogel grippers, research on electromagnetic coupling also exists. These studies effectively demonstrate the precision gripping capabilities of hydrogel grippers under the synergistic effects of electric and magnetic forces. Electromagnetically coupled hydrogel grippers must exhibit both electromechanical and magnetomechanical characteristics to achieve dual responsiveness. During the design and fabrication process, it is essential to explore suitable crosslinking densities and to control the ionic concentration and distribution within the hydrogel. Additionally, the selection and integration of conductive polymers and magnetic nanoparticles require careful consideration. Kim et al. [132] designed an innovative tether-free gripping hydrogel microrobot that can respond simultaneously to electric and magnetic fields. This microrobot features two arms constructed from anodic and cathodic electroactive hydrogels, which bend under the influence of the electric field, thus controlling the “opening” and “closing” of the gripper. Furthermore, the embedded magnetic nanoparticles (MNPs) enable movement within a magnetic field. Under the influence of the electric field, the electroactive hydrogel experiences bending deformation due to the solvent permeation effect resulting from ionic concentration gradients, thus controlling the gripping motion of both arms. This mechanism relies on Gibbs–Donnan equilibrium, causing the cathodic hydrogel to bend towards the cathode and the anodic hydrogel to bend towards the anode, enabling precise control of the gripper (Figure 8f). Moreover, the integrated MNPs can guide the microrobot’s movement under the magnetic field modulation of an electromagnetic actuation (EMA) system. By optimizing the arrangement of MNPs, the sensitivity to turning and propulsion efficiency of the robot are enhanced. Within designated areas, the robot can efficiently turn and advance to target locations by precisely adjusting the magnetic field vector. During operation, the robot demonstrated synchronized execution of grasping and moving tasks. Initially, it opened its arms using an electric field. Then, it approached the target object driven by a magnetic field. Subsequently, a reverse voltage closed the arms to complete the grasp. The magnetic field again propelled the robot and the object to a new position. Finally, the application of voltage released the object. This process achieved precise transportation and illustrated the efficient collaboration between gripping and driving in the robot. This untethered, gripping-type hydrogel microrobot, integrated with electric and magnetic field control systems, opens new avenues in soft robotics research and promises extensive applications in biomedicine and micromanipulation. Furthermore, Jang et al. [134] developed a chitosan hydrogel gripper that combines biocompatibility and biodegradability through 4D printing technology. This gripper features citric-acid-coated SPIONs, enriching the design concepts for electromagnetically driven hydrogel actuators.

Tachibana et al. [133] developed an innovative electromagnetic hydrogel actuator. This actuator assembles liquid metal gel fibers and exhibits ultrafast and highly deformable characteristics. They utilized microfluidic technology to successfully prepare gel fibers with a core–shell structure. The core consists of liquid metal, while the shell is formed by a composite solution of sodium alginate (AlgNa) and PVA. These liquid metal gel fibers assemble into various actuators, such as XY-axis moving platforms, bending grippers, and annular grippers. The core mechanism behind this response lies in the application of Lorentz force (Figure 8g). When the liquid metal gel fibers, positioned in a magnetic field, conduct electricity, the interaction between the current and magnetic field generates a powerful Lorentz force. This force drives the gel fibers and the entire actuator to achieve rapid and precise motion and deformation. Whether translating, rotating, bending, or expanding, these actions can be achieved by precisely controlling the current. In terms of grasping and driving, this electromagnetic hydrogel actuator demonstrates remarkable performance. It can execute complex operational tasks at high speeds and flexibility, such as precisely controlling the displacement of the XY-axis moving platform to achieve accurate object positioning or stably gripping and releasing objects of various shapes and sizes using bending and annular grippers. These features suggest a wide range of applications for this actuator in fields like robotics, micromanipulation, and biomedical engineering. Furthermore, researchers conducted in-depth analysis of the Lorentz force effects and deformation behavior of the gel fibers using COMSOL software and experimentally validated their vibrational characteristics, thereby reinforcing the reliability of the theoretical foundation. However, this actuator also suffers from insufficient material durability, high control complexity, potential electromagnetic safety hazards, and significant energy consumption, warranting further optimization.

In summary, hydrogel actuators demonstrate significant potential in materials science, robotics, and biomedical fields, featuring efficient actuation and shape control under electric fields, magnetic fields, and electromagnetic coupling. However, they face challenges such as long-term stability, biocompatibility, environmental friendliness, and precise control and energy efficiency in complex environments. In the future, technological advancements are likely to address these issues gradually, facilitating the broader application of hydrogel actuators and promoting substantial progress in smart materials and soft robotics technology.

#### 4.2.2. Electronic-/Magnetic-Responsive Grab and Release of Objects

Hydrogel grippers achieve precise deformation control through electric fields, enable remote operation and robust force control via magnetic fields, and enhance rapid response and contactless control by integrating electromagnetic advantages. This greatly expands applications in micro–nano-operations, biomedicine, and cargo handling. This section reviews the latest advancements and broad application about electronic, magnetic, and electromagnetic coupling response.

Electroresponsive hydrogels represent a class of hydrogel materials with unique electrical properties. They can change shape, volume, or stiffness under the influence of electric fields. Such changes typically arise from the rearrangement and migration of ions or molecules within the hydrogel, thus eliciting a response to external electric fields [135]. This technology boasts high driving performance, low energy consumption, and quick response times, making it particularly suitable for applications requiring precise control and rapid reaction. Ko et al. [125] studied an electro-osmosis-driven hydrogel actuator. Their research showed that a planar soft gripper, with a thickness of 500 μm, exhibited slow-release characteristics, allowing it to stably and efficiently grasp and lift objects of varying weights, such as a 20 mg flower (Figure 9a) and a 1000 mg battery (Figure 9b). This hydrogel gripper achieves grasping by applying a positive bias voltage (+3 V), lifts objects in a slow relaxation state (0 V), and releases them through a negative bias voltage (−3 V). Weighing only 32 mg, it can grasp and lift objects approximately 31 times its weight, demonstrating exceptional driving capabilities. This study underscores the significance of biocompatibility and biodegradability in the design of medical robots. By utilizing biocompatible and biodegradable materials, such as magnetic nanoparticles and hydrogel matrices, new avenues emerge for the development of safe and efficient medical robotic platforms. However, while this actuator can function at voltages as low as 0.5 V, the use of copper electrodes limits the bending radius in certain cases. Consequently, actuators of specific thicknesses fail to achieve optimal performance at ±3 V. Overall, electro-osmotic-driven hydrogel grippers exhibit notable advantages in performance, power consumption, strain capacity, lifespan, and energy density. Yet, they still face limitations regarding operational voltage and application scope.

After recognizing the significant advantages and limitations of electrically responsive hydrogel actuators, we turned our attention to another cutting-edge technology, magnetically responsive soft robotic grasping techniques. This technology utilizes the interactions of magnetic nanoparticles, such as SPIONs, within a magnetic field to achieve shape deformation and grasping actions in soft robots [136]. This approach boasts benefits like contactless actuation, fast response speeds, and high control precision, making it particularly suitable for remote control and complex environment operations. Research by Goudu et al. [48] introduced a magnetically driven wireless hydrogel gripper and designed an innovative cargo transfer system that integrates ground rolling motion with magnetic field gradient control to facilitate cargo grasping, transportation, and release. Initially, cargo undergoes transfer through a ground rolling platform. With the assistance of a 25 mT rotating magnetic field, the transportation efficiency is enhanced (Figure 9c). In cases of necessity, the magnetic field gradient generated by permanent magnets endows the robotic arm or magnetic levitation device with the capacity to grasp objects with flexibility, navigate obstacles, and ultimately deliver the cargo precisely to its designated destination. This hydrogel gripper employs the magnetic moment from SPION chains within a magnetic field to bend and deform its gripping arm, effectively completing grasping tasks (Figure 9d). Experimental results show that as the magnetic field strength increased from 5.5 to 25.2 mT, and the bending angle of the gripping arm rose from 3.8° to 58.3°, indicating excellent control performance and grasping capability. However, this gripper also has limitations, such as a reduced payload capacity, which restricts its handling capability. Additionally, its complex manufacturing process, involving permanent magnets and photopolymerization techniques, raises production costs. Magnetic-responsive soft robotics technology exhibits substantial adaptability and application potential across various environments, spanning from terrestrial transport to underwater operations. Tian et al. [129] developed a magnetic soft hydrogel underwater gripper, which integrates shape memory and non-contact bending deformation characteristics, significantly enhancing the engineering application scope of magnetoresponsive hydrogels. The schematic of the magnetic soft hydrogel underwater gripper demonstrates remarkable performance under the combined influence of alternating magnetic fields (AMFs) and driving magnetic fields (Figure 9e). Initially, the gripper remains open and possesses a certain rigidity. Upon applying the AMF, the hydrogel softens due to induced heating, subsequently bending and entangling around the target object. However, the gripper cannot stably lift the object due to reduced stiffness at this stage. Crucially, the removal of the AMF enables the hydrogel to cool, harden, and lock its bent shape, allowing it to securely grasp and lift objects significantly exceeding its own weight (Figure 9f). Experimental evidence has demonstrated that the gripper is capable of securely holding objects up to seven times its own weight. This magnetically responsive shape memory hydrogel has been effectively utilized in a four-armed underwater soft gripper, showcasing its potential for safely and efficiently grasping and lifting heavy objects in complex underwater environments. This strategy not only offers new material options for flexible electronics, soft robotics, and biomedical engineering but also indicates promising prospects for the widespread application of magnetoresponsive hydrogels in future intelligent system construction. Notably, despite its advantages, several challenges remain. These challenges include the necessity of continuous magnetic fields for sustaining deformation, high energy consumption, the demand for increased magnetic field strength to enhance hydrogel rigidity—which complicates both application and cost—and insufficient mechanical robustness, which limits broader deployment in engineering domains. Consequently, further optimization and verification of its practicality are of particular importance.

In addition to the aforementioned applications in micro–nanomanipulation and cargo handling, the magnetic-responsive soft robotic grasping technology demonstrates extensive potential in minimally invasive surgery and targeted drug delivery. The research by Chen et al. [137] introduced an innovative oral small-intestine-targeted drug delivery system using child–parent microrobots (CPMs) as carriers. This system protects and delivers child microrobots under gastric acid conditions, achieving precise drug release in the small intestine. This study not only expands the application of soft robots in drug delivery but also provides new therapeutic strategies for diseases requiring precise targeted treatment, such as neurodegenerative disorders. Further advancing this field, the work by Dong et al. [138] developed a highly integrated multifunctional soft helical microswimmer capable of targeting neuronal cells for delivery and in situ stimulation, opening new pathways for cell therapy in central nervous system injuries and diseases.

Electromagnetic-responsive soft robotic grasping technology combines the benefits of electric and magnetic fields, achieving more flexible and efficient grasping operations through integrated driving elements. These grippers typically feature multiple degrees of freedom, high adaptability, and precise control capabilities, demonstrating broad application prospects in micro/mm-scale object manipulation, biomedical fields, and flexible electronics [139]. Kim et al. [132] investigated a hydrogel microrobot that employs a combination of electroactive hydrogels and magnetic nanoparticles (MNPs) for dual-drive capabilities via electric and magnetic fields. The electroactive hydrogels, synthesized through ultraviolet polymerization, exhibit bending behaviors toward the cathode and anode under electric fields, enabling open and close motions. Additionally, an external magnetic field allows for precise manipulation of hydrogel microrobots via the EMA system, facilitating alignment and directional adjustments. Initially, the hydrogel microrobots enter a preparation phase, assuming an open posture to receive and grasp target objects. Subsequently, under the influence of a precisely applied external magnetic field, the microrobots navigate to the target’s location to execute the grasping action. Upon nearing the target, the application of an electric field prompts the microrobots to swiftly close their structures around the object, securing it tightly. Once the object is successfully grasped, the robot gains the ability to move flexibly forward or backward while maintaining a firm grip through magnetic field control. Finally, to release the object at the designated position, the microrobots perform an opening motion by reapplying the electric field, gradually loosening their grip until complete release, effectively completing the operation (Figure 9g). These hydrogel microrobots, leveraging dual driving mechanisms, demonstrate precise control and complex movements with exceptional compatibility for soft, smooth, and fragile objects. They also feature autonomous water absorption recovery in liquid environments to ensure reusability. However, the production of electrode residues and the necessity for electrolyte cleanliness pose challenges in practical applications, affecting operational efficiency and cost-effectiveness, necessitating improvements in electrode materials and strategies to mitigate debris impact. Tachibana et al. [133] designed a hydrogel gripper that harnesses Lorentz forces for rapid actuation and large deformations. By passing a current through coiled or linear gel fibers, the system generates a magnetic field that interacts with the current within the fibers, resulting in deformation. The coiled fibers exhibit greater deformability owing to variations in pitch. The gripping images illustrate the practicality of the ultrasoft ring gripper for transporting hydrogel blocks (Figure 9h). In the experimental setup, neodymium magnets were strategically positioned at three locations. Energizing these positions allowed the gripper to expand and perform grasping and releasing actions. Initially, the gripper securely grasped a blue hydrogel object at position (i) before moving to position (ii) to release it. Subsequently, without altering its configuration, the gripper successfully grasped a red hydrogel object again at position (i) and ultimately moved to position (iii) to release it. This process not only confirmed the gripper’s excellent grasping ability for soft, smooth hydrogel materials but also highlighted its reusability and multitasking capabilities, demonstrating significant potential in micromanipulation. Additionally, the gripper exhibited rapid response in vibration tests and demonstrated notable deformation speeds, indicating promise for high-frequency operations. The ultrasoft contact properties of this material render it particularly well-suited for the manipulation of delicate objects, such as human tissues and cells. This characteristic suggests significant applications in biomedicine, precision manufacturing, and food processing. Similarly, Merhebi et al. [140] demonstrated that combining liquid metal nanoparticles with magnetic nanoparticles resulted in conductive and magnetically responsive gels, paving new paths for the multifunctionality of soft robots.

In summary, technologies involving electrical, magnetic, and electromagnetic-responsive hydrogel grippers have made significant strides in the realm of soft robotic grasping. Each type displays distinct advantages, and electrically responsive hydrogels offer high actuation performance, low energy consumption, and rapid response capabilities, supporting precise control and efficient grasping. Magnetic-responsive soft robots demonstrate exceptional performance through non-contact remote manipulation, particularly in complex environments and cargo handling. Meanwhile, electromagnetic-responsive soft actuators cleverly combine the benefits of both, enabling more flexible and effective grasping operations. However, these technologies also face challenges, such as the need for improved biocompatibility in electrically responsive hydrogels, stability issues of magnetic systems in intricate electromagnetic environments, and the complexity of control in electromagnetic fusion technologies. Continuous advancements in materials science, control theory, and micro–nanotechnology are poised to significantly enhance innovations in soft robotics. These developments will create a robust foundation for the widespread application of soft robotics in micro–nano-operations, biomedicine, and cargo handling. Furthermore, this progress is likely to stimulate additional innovation and development across related fields.

### 4.3. Chemical-Responsive Hydrogel Grippers

#### 4.3.1. Chemical-Responsive Driving Mechanisms

Stimulation-reactive hydrogels are highly responsive to various chemical stimuli. Under the action of external stimuli, these hydrogels show a series of specific behaviors, including expansion, contraction, bending, deformation, and dehydration [102]. These unique properties make stimulus-responsive hydrogels an ideal choice for manufacturing grippers. This section will comprehensively discuss the latest research progress of the gripper based on chemical-stimulus-responsive hydrogel, covering pH response, dual response to pH and temperature, ionic response, and electrochemical response [141].

A hydrogel gripper which responds to pH usually contains side groups that can be ionized in solution on its polymer skeleton, which can be divided into anionic side groups and cationic side groups according to the types of charges after ionization. When the pH value is higher than the degree of acid dissociation (PKa), the side-chain groups dissociate, and the hydrogel loses a proton and is negatively charged after the side-chain groups dissociate, which is an anionic side group. At this time, the osmotic pressure in the hydrogel network decreases, and the hydrogel expands driven by the osmotic pressure, while when the pH value is lower than PKa, the hydrogel with anionic side groups contracts. On the contrary, cationic pendant hydrogels swell when pH is lower than PKa and contract when pH is higher than PKa. Based on this, the clamping behavior of pH-responsive hydrogels was studied. The pH-responsive hydrogels prepared by Shan et al. showed significant swelling differences in different pH environments [142]. The swelling rate of hydrogel is the lowest in the environment of pH 7, while it reaches the highest in the environment of pH 12 (Figure 10a). The mechanism behind this different swelling behavior is mainly attributed to the electrostatic interaction between polymer chains. Specifically, at pH 2, the amino group on the DMAEMA unit is protonated, while at pH 12, the carboxyl group on the AA unit is dissociated into negatively charged carboxylic acid, which leads to electrostatic repulsion between polymer chains (Figure 10b). This electrostatic repulsion enhances the osmotic pressure in the hydrogel network, thus promoting the swelling, especially when the content of AA units in PAD exceeds that of DMAEMA units, with high swelling rate at pH 12 (Figure 10c). At pH 7, the positively charged protonated amine on DMAEMA and negatively charged carboxylate on AA create an electrostatic attraction that reduces the hydrogel’s swelling, yielding the lowest swelling rate. This pH-induced asymmetry enhances the hydrogel’s rapid bending response, allowing for quick object grasping. More interestingly, PAD4 hydrogel can exhibit complex deformation behavior under different pH conditions and can recover to its original shape in ethanol (Figure 10d). This is due to the high hydrophilicity of ethanol, which can remove water molecules from hydrogel and restore its shape. The excellent shape recovery performance of this anisotropic hydrogel enables it to quickly release the grabbed object. Shu et al. [143] successfully prepared a dual-responsive gripper composed of a bilayer structure of poly(acrylamide-acrylic acid-3-acrylamidophenylboronicacid)/poly(N-isopropylacrylamide) (P(AAm-AAc-3-AAPBA)/PNIPAM), which responds to both pH and temperature stimuli. As shown Figure 10e, the critical solution temperature of PNIPAM is low (LCST is about 32 °C). PNIPAM chains shrink through intra-chain hydrogen bonding above the LCST and expand by interacting with water below it, while AAc exhibits reversible pH responsiveness near its PKa of 4.25. When the pH value is greater than PKa, P(AAm-AAc-3-AAPBA) expands; when the pH value is less than PKa, P(AAm-AAc-3-AAPBA) also swells. With the increase in temperature, the double-layer structure bends in the opposite direction and the angle increases, which is attributed to the contraction of the P(AAm-AAc-3-AAPBA) layer and caused by the strong hydrophobicity of 3-AAPBA. When the hydrophobic force of the PNIPAM layer is greater than that of hydrogel, P(AAm-AAc-3-AAPBA)/PNIPAM will be deformed in reverse bending, and the water absorption capacity of PNIPAM will be weakened and the reverse bending angle will be increased when the temperature rises again. The AAM-AAC-3-AAPBA/PNIPAM double-layer structure has reversible bending behavior in low-temperature acidic and high-temperature alkaline environments, which can be accurately controlled by adjusting pH value and temperature (Figure 10f).

There is also an ion-responsive hydrogel gripper, which expands and contracts mainly according to the variety and concentration of ions, and the core is attributed to the change in osmotic pressure in hydrogel matrix. The state of ion response can be adjusted according to the addition of different types and concentrations of ions. Yang et al. [144] found that hydroxide produced at the cathode during water electrolysis catalyzes the covalent crosslinking of CS and ECH, and the covalent crosslinking area (CS-ECH) of the hydrogel is controlled by electrowriting (Figure 10g). The rigid chain network formed by covalent crosslinking in the written area did not show significant bending deformation when further soaked in SDS solution, but the CS hydrogel in the unwritten area did, which made the CS-SDS hydrogel have bidirectional permeability. As shown in Figure 10h, with the change in ion concentration in the aqueous solution, H_2_O can pass through the shell, thus balancing the osmotic pressure between the outside and the inside of the hydrogel. In deionized water, the inflow of H^+^ causes it to swell. In NaCl solution, there is the opposite effect to H^+^ outflow, which leads to shrinkage due to dehydration, and then clamping occurs. In addition, the swelling ratio curve of CS−SDS hydrogel switching between deionized water (10 min) and 2M sodium chloride solution (10 min) is shown (Figure 10i). CS–SDS hydrogel has shape memory effect and good fatigue resistance in solutions with different ion concentrations, which shows that C–SDS hydrogel has high reliability and mechanical stability when used in a gripper. Meanwhile, Su et al. [147] developed pH-responsive grippers and clamps with a high load-to-weight ratio using the stimulus-responsive properties of hydrogels. The latest research shows that CS−SDS hydrogel has shape memory effect in solutions with different ion concentrations [145]. As shown in Figure 10j, when the gelatin core is mixed into an alginate solution by soaking in Ca^2+^, an alginate layer with ionic strength response will be formed along the gelatin core through alginate–Ca^2+^ crosslinking. Since the response of the alginate layer is limited, another thermally responsive alginate-poly (2-(dimethylamino) ethyl methacrylate) (Alg-PDMAEMA) layer is subsequently introduced to the double-layer structure and was studied by a scanning electron microscope. With the introduction of the PDMAEMA network, the negative charge and positive charge of the Alg-PDMAEMA layer will shield each other, and the Alg-PDMAEMA network will not swell in deionized water and NaCl solution. Therefore, if the Alg/Alg-PDMAEMA double-layer hydrogel is immersed in deionized water, the hydrogel will bend to the alginate -PDMAEMA layer, and when it is transferred to 0.1 m NaCl solution, the hydrogel will bend to the alginate PDMAEMA layer to complete the clamping action (Figure 10k). By changing the ambient temperature and ionic strength, the precise control of this hydrogel actuator can be realized. At low temperature and low ionic strength, the hydrogel actuator keeps shrinking. At high temperature and high ionic strength, the hydrogel actuator expands.

In addition, there are hydrogel grippers with electrochemical response. When preparing such grippers, considerations must include designing effective ion transport channels, incorporating chemical groups for redox reactions, selecting appropriate crosslinking networks, ensuring optimal hydrogel–electrode interface contact, and enhancing electron transfer efficiency. Zhu et al. [146] prepared a chitosan hydrogel using anodic electrodeposition, which can generate H^+^ and introduce Fe^3+^ into the chitosan hydrogel, endowing it with responsiveness to ion concentrations in the aqueous environment. As shown Figure 10l, the charged gripper absorbs counter ions in sodium chloride solution to form an electrostatic shielding effect and contract. The density of Fe^3+^ complexes at the top of the writing area is high, and the electrostatic shielding effect is stronger in salt solution, which leads to the high deflation rate of the gripper. The protonation of chitosan on the surface of the writing area makes it swell in deionized water, which drives the driver to bend to the bottom, and it bends to the top in sodium chloride solution, which corresponds to the swelling behavior. In sodium chloride solution, the actuator bends to the top of the writing area with high complex density. The bending angle increases with the increase in NaCl concentration (Figure 10m). In high-concentration NaCl solution, Fe has a stronger electrostatic shielding effect, larger shrinkage difference, faster bending rate, and larger bending angle (110 at 1 M NaCl). Due to the electrostatic shielding effect, the bending angle increases with the increase in soaking time. When the concentration of NaCl solution is low (0.1 M), the electrostatic shielding effect is weak, the swelling rate is reduced, and the bending angle is reduced. These results show that the Fe^3+^-complexed chitosan hydrogel actuator has ion responsiveness and can form different deformations in response to different ion concentrations (Figure 10n). Because of its high deswelling rate in salt solution, it can improve the stability when gripping objects.

To sum up, the chemical stimulus response hydrogel gripper has been widely explored in scientific research and daily life. It can lift objects up to 500 times its own weight, and the double-layer structure of P(AAm-AAc-3-AAPBA)/PNIPAM can respond to temperature and pH value cooperatively [148]. The bending angle and speed are greatly improved, and the protonation of ion-responsive chitosan hydrogel on the surface of the writing area increases its expansion in deionized water, and the bending angle increases with the increase in NaCl concentration, thus achieving stable operation in various grasping modes. In the future, the hydrogel gripper can be designed as a micromedical device, which can perform complex tasks or perform remote task operations in organisms [149]. Hydrogel can be combined with artificial intelligence and machine learning to realize intelligence and mechanization [150]. We believe that the development of chemically stimulated hydrogel grippers will proceed rapidly.

#### 4.3.2. Chemical-Responsive Grab and Release of Objects

Most natural drives, such as muscle-driven movements and the folding of mimosa, are basically derived from the subtle anisotropic structures naturally developed in organisms. In recent years, bionic synthetic actuators have attracted extensive research interests because of their scientific significance and potential applications, including artificial muscles [151], soft robotics [71], sensors, and imitation equipment [152]. Among them, polymer hydrogels, widely known as soft and wet materials, have attracted special attention because of their bionic characteristics and fascinating response characteristics when exposed to external stimuli [153]. This section will comprehensively discuss the latest research progress of the gripper based on chemical-stimulus-responsive hydrogel, covering pH response, dual response of PH and temperature, ionic response and electrochemical response.

The grasping and releasing of objects by hydrogel grippers in response to pH value have been studied. Le et al. [154] developed an innovative method for preparing a double-layer hydrogel actuator, which utilizes polyvinyl alcohol (PVA) polymer chains to form asymmetric crystal regions within a chitosan matrix prepared from an alkaline solution. Through this strategy, chitosan/PVA composite double-layer hydrogel was successfully synthesized, which showed excellent pH-responsive bending behavior and mechanical strength, and could be used as an efficient pH control driver (Figure 11a). In this study, researchers designed a bionic manipulator, whose structure imitated the shape of the letter “X” to simulate the ability to grasp heavy objects. The manipulator consists of four arm-shaped double-layer hydrogels, the ends of which are fixed on metal wires. In the experiment, a 3.0 g spherical object was placed in an acidic aqueous solution with a pH value of 2. The H-Gel layer of the manipulator bent the whole structure within 2 min due to water absorption and expansion and successfully grabbed the heavy object and lifted it from the solution. Subsequently, the manipulator was placed in an alkaline environment and quickly returned to its original shape and released the object. This process shows the quick response and deformation ability of an intelligent manipulator to perform basic mechanical tasks under extremely acidic conditions. Jiang et al. [143] explored the application of double-layer structures by demonstrating with bionic hands, where the fingers were tailored to different lengths. A soft, translucent plastic block is chosen as the palm part. The palm and fingers are glued into a bionic hand and tested (Figure 11b). When the temperature drops to 10 °C and the pH value is adjusted to 2, the manipulator bends gradually, simulating the gripping action. Then, it is transferred to a solution with a temperature of 55 °C and a pH value of 12, at which time the fingers of the manipulator unfold rapidly. Additionally, we can observe changes in the finger color: at lower temperatures, the PNIPAM layer is transparent and hydrophilic, while at higher temperatures it is in an opaque, hydrophobic white state. This change is due to the change in hydrophilic and hydrophobic properties of the PNIPAM layer at different temperatures.

Hydrogel grippers with variable shape and wide adaptability are needed in many fields, such as intelligent sensors, circuit switches, and soft clamps [155]. Yu et al. [156] developed a double-layer hydrogel gripper composed of PNIPAM and PDMAEMA. The gripper can realize directional bending according to the change in temperature and pH value, so it can be used as an environmentally responsive actuator. We built a four-arm intelligent manipulator, whose shape can be adjusted with the change in environmental temperature and pH value, and then demonstrated its ability to grasp black rubber blocks. The size of each robot arm is precisely 12 mm in length and 5 mm in width to ensure its effectiveness as a pH response system (Figure 11c). First, the double-layer hydrogel gripper is placed in 0.1 M HCl acid solution to grab the rubber block, and then it is transferred to ethanol solution to reduce the weight. This process demonstrates the ability of soft actuators to hold and release heavy objects. In an acidic environment, the PDMAEMA layer acts as a positive electrode to promote the clamping of rubber blocks, while in ethanol the interaction between polymer and medium is weakened, which leads to the contraction of the PDMAEMA layer and the weight reduction. It is worth noting that the double-layer actuator can quickly respond to pH changes, complete grasping in about 35 s, and release in about 20 s. In addition, due to its thermal responsiveness, the gripper can also perform a grasping task in high-temperature water. Immersing the four-arm hydrogel gripper in deionized water at 45 °C triggered it to bend towards the PNIPAM layer, because the PNIPAM layer would shrink when the temperature was raised, thus realizing rapid grasping as a positive layer. The gripper can quickly capture an object in about 15 s. When transferred to cold water at 15 °C, the gripper can release the rubber block it grabs within 30 s. This double-layer gripper can lift an object nearly five times its own weight and can effectively grasp it in the environment with temperature and pH changes. This intelligent double-layer hydrogel is expected to play an important role in many application scenarios such as hull cleaning, wastewater treatment, and drug delivery because of its strong adaptability. Zhou et al. [145] immersed gelatin cores in Ca^2+^, resulting in the formation of an ion-intensity-responsive alginate layer along the gelatin cores, which was crosslinked by Ca^2+^. Subsequently, another thermoresponsive layer of Alg-PDMAEMA was introduced, creating a double-layer hydrogel scaffold with both ion intensity and temperature responsiveness. As shown Figure 11d, when the hollow hydrogel sphere is transferred from 0.1 M NaCl solution to deionized water, the alginate layer inside it will swell due to water absorption, resulting in the opening of the whole hydrogel structure. When the temperature rises to 60 °C, the outer Alg-PDMAEMA layer of hydrogel will shrink due to thermal response, and the capsule shape will expand. This shape change caused by ionic strength and temperature is reversible. These hydrogel capsules with programmable shape change ability show potential application prospects in the field of drug delivery. They can control the release of drugs according to the ionic strength and temperature changes in the environment.

**Figure 11 biomimetics-09-00585-f011:**
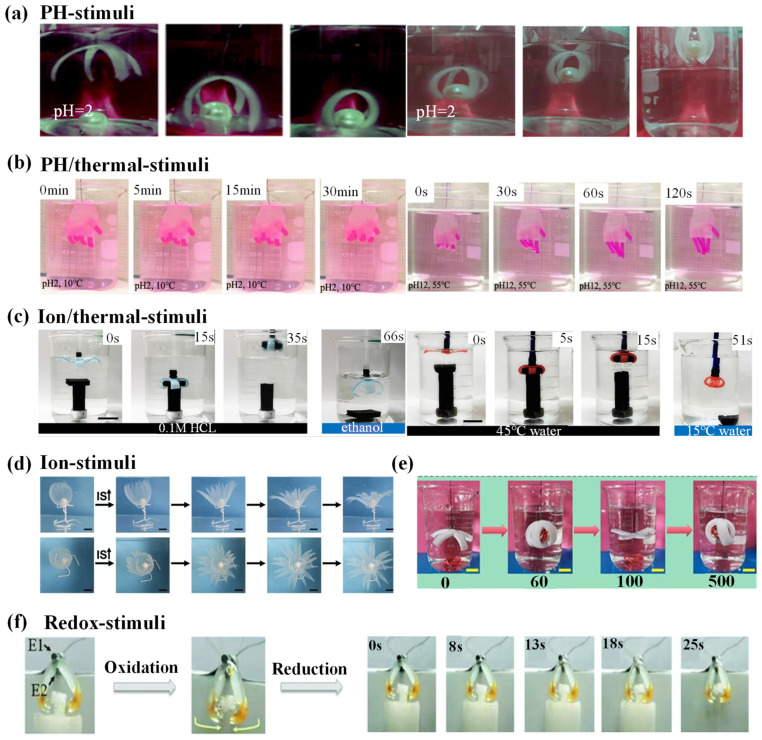
(**a**) The action of grasping an item using an “X”-shaped gripper within an acidic solution with a pH level of 2 [154], Copyright 2022, the Royal Society of Chemistry. (**b**) The act of a prosthetic hand mimicking the motion of clenching and then extending its fingers [143], Copyright 2023, Published by Elsevier Ltd. (**c**) The weight is captured by a bilayer hydrogel as a four-arm gripper in response to pH and temperature as shown in schematic illustration photographs [156], Copyright 2017, Elsevier B.V. (**d**) The hollow hydrogel capsules are subject to the ionic strength and thermoresponsive shape deformation process [145], Copyright 2020, WILEY-VCH Verlag GmbH & Co. (**e**) Schematic illustrations and images of cross-shaped bilayer hydrogel actuating behaviors powered by ionic fuels [157], Copyright 2024, Elsevier Inc. (**f**) Hydrogel strip with ion printed ends lifts polystyrene foam with positive potential on E2 and releases it with negative potential. Actuating clams by applying oxidative potential [158], Copyright 2020, the Royal Society of Chemistry.

In addition, new smart materials show broad application prospects, such as hollow hydrogel capsules prepared from spherical gelatin cores, which can expand in response to changes in ionic strength and temperature, making them promising candidates in the field of drug release. We believe that the current research will promote the design and manufacture of new intelligent materials with wide potential applications. In the field of autonomous soft robots, hydrogel actuators driven by chemical fuels play a key role. However, the accumulation of chemical wastes generated by chemical fuels limits the further development of programmable and reusable hydrogel drive systems. Zhao et al. [157] proposed the concept of ion-fuel-driven soft robots. The soft hydrogel actuators were developed by coordinating the swelling and shrinking abilities of Janus double-layer hydrogels, powered by decomposable and easily removable ion fuels. Adding 0.6 M (NH_4_)_2_CO_3_ into deionized water as the fuel of the gripper can realize the clamping and releasing of the object. As shown Figure 11e, the design of the cross-shaped double-layer hydrogel gripper driven by ion fuel shows its response ability to the change in (NH_4_)_2_CO_3_ ion intensity and can realize the clamping and releasing actions. By adding 0.6 M (NH_4_)_2_CO_3_ into deionized water as the driving fuel, the double-layer hydrogel gripper showed opposite expansion and contraction behaviors during operation, resulting in strain mismatch between the two layers, which led to the expansion of one layer and the contraction of the other layer, forming a clamping action. The direct decomposition of ammonium carbonate gives the system the ability of self-driving, and it can spontaneously complete the function of grasping and releasing objects without external stimulation.

Fast response and reversible shape memory characteristics are very important for bionic soft actuators made of polymer hydrogels. However, many traditional hydrogel actuators are usually slow in driving speed or deformation ability in water. Especially for the rapid actuation of hydrogel in air, it is still a difficult problem to be overcome. Yang et al. [158] developed stretchable polyacrylamide nanocomposite hydrogels by precisely controlling local crosslinking through coordination with multivalent metal ions such as Fe^3+^, using redox-responsive hydrogels. The redox reaction of Fe^3+^ reversibly regulates local coordination and crosslinking, so that the hydrogel can respond quickly. The reversible conversion between Fe^3+^ and Fe^2+^ can be realized by applying oxidation or reduction bias on the iron electrode. The chelating ability of Fe^3+^ ions with amino groups is stronger than that of Fe^2+^, so when Fe^3+^ is reduced to Fe^2+^, the local crosslinking density will be reduced, resulting in the deformation of the hydrogel. Two hooks were formed on PAAM/MMT hydrogel by ion printing technology, and then the hydrogel was placed between a graphite electrode and iron electrode. As shown Figure 11f, when an oxidation bias (10 V) is applied to the iron electrode, Fe^3+^ ions are released, and the hydrogel quickly bends towards the electrode, and the hook closes to catch the object. When the reduction bias (−10 V) is applied, the hydrogel arm moves outward and releases the object quickly. The whole process is reversible and responsive, and it bends and grabs in about 20 s after oxidation and opens and releases in about 100 s, which shows the rapid reversible driving ability of hydrogel actuator. With the appearance of soft robot technology based on hydrogel [159], researchers have demonstrated the potential of hydrogel-based robotic components [160]. However, the efforts to combine various components into a system are still in the initial stage. SAA hydrogel shows excellent ability in human motion detection and physiological signal response [135]. Hydrogels show excellent recognition ability for various stimuli (such as temperature and pressure) and can easily recognize different stimuli by the waveforms of electronic signals and quantify the intensity of stimuli by the response-recovery period (RRC). These characteristics make it an ideal choice to develop high-performance multifunctional bionic skin based on ionic conductive hydrogel with a low-cost zwitterionic structure. Although the potential of hydrogels as soft robot materials has been preliminarily verified, this field is still in its infancy. We look forward to continuous research efforts to promote this kind of soft robot to be more widely used in daily life in the future.

### 4.4. Others

#### 4.4.1. Other Driving Mechanisms

In addition to the aforementioned common thermally, electromagnetically, and chemically responsive hydrogel grippers, researchers have also developed hydrogel grippers with other external response modes, including moisture, DNA, and ultrasound. Compared to single- or double-response hydrogel grippers, multiresponse hydrogel grippers will be a development trend due to their broader application prospects in the future, and we will underscore the significant research of multiresponse hydrogel grippers in terms of actuating mechanisms at the end of this section.

As the source of life, water can be used as the condition to trigger hydrogel gripper actuators as well. Xue et al. [161] proposed a constitutive model of water-triggered shape memory hydrogels to successfully fabricate bilayer and four-arm structure grippers. The keys to the water trigger are the frozen deformation gradient and dense phase volume fraction of hydrogels. The water content of hydrogel triggers the shape memory process, for example, when the water content decreases, the sparse phase transforms into the dense phase. The pre-deformation locks in the dense phase under constraint, **F_d_** in the dense phase is released, and **F_f_** is retained. As the hydrogel is hydrated to a high water content, **F_f_** is released, and the temporary shape reverts to its initial shape (Figure 12a). The relationship between the deformation gradients in the two phases can be expressed as **F = F_s_ = F_d_F_f_**. In addition, taking the total deformation gradient of the polymer network and the chemical potential of the solvent as independent variables, the expressions of the free energy function and the Cauchy stress are derived. A thermodynamic constitutive model of water-triggered hydrogels based on the assumption that the hydrogel is a mixture of dense and sparse phases is developed, and two internal state variables are used to describe the shape memory effect. Another work focuses on the structure design to achieve moisture-driven hydrogel grippers. Li et al. [162] fabricated a tri-layer actuator named YYI including “Yin” (PET), adhesion (PEA), and “Yang” (PAM) layers. PET and PAM are the layers responding to moisture to actuate, and PEA as an adhesive that makes “Yin” and “Yang” stick together. The mechanism of YYI to fabricate moisture-responsive hydrogel grippers is that PAM hydrogels containing some water and hydrophilic amino groups are sensitive to moisture and exhibit rapid absorption and desorption of water molecules. In contrast, PET, PVDF, PE, and PI polymer films are inert to moisture and do not undergo obvious volume changes (Figure 12b). However, the actuation is only maintained for one to two cycles, mainly due to the plasticization of water molecules in the PAM hydrogel film. The experimental results show that the actuator can bend to 405° in approximately 8 s and return to its initial state within 6 s, showing stability over 1000 cycles under a relative humidity difference of 60%.

Increasing the degree of swelling of DNA is an essential science problem to achieve DNA-triggered shape-changing hydrogels. Cangialosi et al. [163] proposed that swelling would be enhanced if experimenters continuously lengthened crosslinks by using a DNA hybridization cascade in which multiple DNA molecules are inserted into a duplex. They proved the proposed theory by designing DNA sequences consisting of hydrogel crosslinks and corresponding H1 and H2 for the cascade (Figure 12c).

In the research of ultrasound response, Son et al. proposed a hybrid gripper that uses ultrasonic energy as an external trigger system to perform pre-programmed shape transformation in invisible and non-selective environments. The hybrid gripper is composed of three different types of hydrogels, including an ultrasound-responsive NIPAM-based gel, a non-stimulus-responsive structural AAm-based gel, and a magnetically responsive ferrofluid gel. The NIPAM-based gel achieves swelling and contraction through temperature-sensitive changes in crosslinking density under ultrasound, and the layered silicate provides shear thinning properties and acts as an ultrasound scattering site. The AAm-based gel provides structural support and piezoelectric bimorph actuation, and the ferrofluid gel is used for the precise manipulation and movement of the gripper (Figure 12d). As a whole, the key core of ultrasound response is multifunctional layer/elements combined together.

At the end of this section, we want to discuss multiresponse hydrogel grippers. As described above, the key core to fabricating multiresponsive hydrogels grippers is: (i) clear hydrogel gripper research objectives and applicable fields. (ii) Find hydrogels or other functional polymer/particles for targeted response conditions. (iii) Clarify the selected material’s stimuli-responsive mechanism. (iv) Explore various parts’ combination mode from physical, chemical, or physicochemical coupling. (v) Determine fabrication strategies to achieve grippers’ preparation. (vi) Optimize materials ratio and structure design to obtain excellent performance. Multistimuli-responsive hydrogel grippers are an effective combination of individual part responsive functions. The rationality of mechanism description is the premise and the key to the success of hydrogel gripper preparation. Multiresponse hydrogel grippers are also a popular research direction at present. For example, Lu et al. [164] prepared new programmed hydrogels which can be triggered by pH, temperature, and light and can be reshaped using lighting. They introduced the partially reversible light-directed assembly (PRLDA) method by combining the spatiotemporal control of light and the photoreaction of coumarin, using microgels (MGs) as building blocks to construct multiresponsive hydrogels. The hydrogels’ pH and temperature responsiveness originated from the MAA and MEO_2_MA segments (pH-triggered hydrogen bond formation between protonated -COOH groups within MAA), and the light responsiveness originated from the CMA unit after optimization. The programmed hydrogel grippers have potential for biomaterial and optoelectronic applications.

We believe that when achieving multifunctional characteristics and expanding the application fields, multiresponse hydrogel grippers will have a promising future in actual applications in future research.

#### 4.4.2. Other Grab and Release of Objects

In addition to the previously mentioned common forms of hydrogel grippers that respond to the external environment for grasping and releasing objects, this section also presents research on the combination of experiments and simulations in the domain of hydrogel grippers. It also covers the grasping and release of objects with other responses such as to light, moisture, DNA, and ultrasound.

Xue et al. [161] proposed the hypothesis that hydrogels consist of both dense and sparse phases. A thermodynamic constitutive model of water-triggered shape memory hydrogels (SMHs) has been established in this study. In the proposed model, two internal state variables are utilized to describe the shape memory effect. The first internal state variable is the frozen deformation gradient, used to analyze the deformation stored and released during the shape memory and shape recovery processes, respectively. The second internal state variable is the volume fraction of the dense phase in the phase transition process, quantifying the degree of phase transition caused by hydration and dehydration. To further validate the accuracy of the model in describing shape memory behavior, experiments and simulations were conducted on grasping and release processes of a gripper structure. The simulation results and experimental results are shown Figure 13a. Through comparing simulations and experiments for different shapes in the shape memory cycle, it was found that there was good consistency, validating that this model can predict water-triggered shape memory behavior of hydrogels under complex deformations. Additionally, to verify prediction ability, a self-bending process was simulated using the finite element method (FEM) with encoded UEL. Comparison of simulation results and experimental results can be observed in Figure 13b. The study revealed that the model effectively explained the self-bending shapes observed in double-layer structures, indicating its ability to describe the process of water-induced shape recovery in hydrogels. This finding aligns well with academic standards and contributes to a better understanding of hydrogel behavior. These research findings offer researchers a theoretical framework for studying mechanical behavior of water-triggered SMHs as well as providing a tool for designing new structures based on SMH optimization purposes within academic parameters.

Regarding the grasping and release of objects corresponding to the response to ultrasound, Yoon et al. [165] proposed the use of dual biological 3D printing technology to successfully manufacture a hybrid hydrogel gripper (Figure 13c). This gripper is capable of selectively picking up and placing targets through external unrestricted control in a dynamic and chaotic environment. To demonstrate its functionality, the hybrid hydrogel gripper was applied to complete a pick-and-place task, navigating through a maze (Figure 13d). By integrating unconstrained ultrasonic drive and magnetic motion control, salmon roe was successfully retrieved from the maze. The hybrid gripper employs magnetic guidance to approach the target, utilizes an ultrasonic opening mode for grasping the target, then uses magnetic guidance to transport the target to the specified location, and ultimately employs an ultrasonic closing mode for releasing the target. In order to successfully transport salmon roe out of the maze, the hybrid gripper must not only undergo ultrasonic-triggered shape transformation from an open to a closed tip but also possess unconstrained control over movements such as rotation, direction change, and linear movement. Initially, the gripper is positioned at the entrance of the maze in an open-legged state. Subsequently, it uses an external magnet for guidance to reach the salmon roe located at the center of the maze and grabs it through low-intensity ultrasonic drive. Once secured, the gripper, with the salmon roe clamped, is guided out of the maze. The fish eggs are released from the maze when its legs return to their open state. Due to its compliance, there is no alteration in the appearance of the salmon roe during this experiment. To complete the pick-and-place task, the hybrid hydrogel gripper navigates through a maze and extract salmon eggs by integrating wireless ultrasonic drive and magnetic motion control. In the experiment, the gripper was capable of achieving wireless motion control, including rotation, changing direction, and linear movement, as well as shape transformation triggered by low-intensity ultrasound. However, it is important to note that the attenuation coefficient of hydrogel is significantly larger than that of biological tissue, being approximately five times that of muscle. This imposes a limitation on the upper limit of available ultrasonic intensity. The high attenuation coefficient can be partly attributed to the printability of hydrogel. Therefore, further optimization for printing more viscous hydrogels is necessary for future research endeavors. Additionally, the time constant of the grasping drive is relatively slow. Further refinement of the hydrogel composition or the utilization of focused ultrasound may be required to address this issue in a scholarly paper.

In addition, Cangialosi et al. [163] proposed that specific DNA molecules can significantly increase the volume of hydrogels by up to 100 times through continuous expansion of the crosslinking points, as demonstrated through research. Utilizing photolithography technology, the researchers were able to integrate multiple regions into centimeter-sized gels, with each region capable of producing distinct shape changes based on different DNA sequences (Figure 13e). In the experiment, DNA-crosslinked polyacrylamide hydrogels were utilized. DNA sequences containing hydrogel crosslinking points and corresponding hairpin structures were specifically designed for cascade reactions. At the same time, a lithography process was developed to accurately define the shape of multimaterial DNA hydrogels, effectively addressing issues such as low modulus, easy adhesion, and potential damage to DNA from ultraviolet rays. The relative concentrations of polymerized hairpins and terminating hairpins can be adjusted to control the expansion of the gel to a predetermined final size. Additionally, effective control of expansion in multiple regions is achieved through the design of different DNA sequences. Specific biomolecular signals can determine the situation and degree of shape change in specific areas of synthetic materials. This DNA-based hydrogel demonstrates a responsive ability to specific DNA trigger signals, thus enabling complex and programmable shape changes. In the study, researchers successfully engineered structures such as petals and “crab” devices, validating that multiple DNA-sequence-responsive hydrogels can undergo shape transformation in response to various hairpin inputs. Furthermore, these structures were found to maintain their altered shapes for at least 60 days.

Next is the research on response to environmental humidity. Li et al. [162] proposed the development of a moisture-driven three-layer YYI actuator (Figure 13f) that demonstrates significant bending curvature and rapid response speed when exposed to moisture. It can achieve rapid jump drive, as well as large-amplitude oscillation with a small temperature change, showcasing multideformation behavior. This actuator has potential applications such as smart switches, mechanical grippers, and crawling and jumping actuators. A moisture-controlled electrical switch, utilizing the reversible driving response, was developed. When the relative humidity (RH) exceeds 60%, the YYI film initially covers the sensor. The circuit is linked to four light-emitting diodes (LEDs) operating under a direct current voltage of 8.37 V. Upon application of moisture to the film in the tube, it unbends and moves away from the sensor within 2 s, causing the circuit to disconnect and turning off the LED. Subsequently, as moisture is removed, the film gradually bends back to cover the sensor within approximately 8 s and reconnects the circuit. This rapid oscillating motion permits quick switching of LEDs on and off at a frequency of up to 1.6 Hz. Nevertheless, due to its relatively slow frequency, further research is required to enhance response speed by increasing ΔRH or utilizing alternative materials with higher moisture sensitivity. For the mechanical gripper, the initial environmental relative humidity (RH) is approximately 60%. The actuator is initially in a stretched state. As the actuator gradually enters a dry beaker with lower humidity, it bends and successfully grasps a piece of foam weighing up to 0.079 g within 1.20 s. The foam is then transferred back to the environment within 2.32 s and released in about 0.4 s. It should be noted that the maximum weightlifting ratio (m/m_0_) achieved by the gripper is approximately 23, where m_0_ and m represent the masses of the actuator and the foam, respectively. The grasping time of 1.20 s aligns with that of most humidity-responsive grippers at present. However, both the grasping time and weightlifting ratio of this specific gripper do not fully meet current requirements. Future work should therefore prioritize improving mechanical performance and response speed through further exploration and manufacturing of highly sensitive humidity-responsive materials. For the crawling actuator, it is placed in an environment with a relatively high humidity of about 60%. When a significant amount of CaCl_2_ desiccant is placed near the film, the loss of moisture in the PAM hydrogel layer causes the film to bend and form an arch, resulting in less friction on the hind legs (F_1_) than on the front legs (F_2_). Similarly, when the desiccant is removed, the friction on the hind legs (F_1′_) becomes greater than that on the front legs (F_2′_), leading to continuous movement on a flat and rough substrate. The arched film can move forward by approximately 9 mm at a speed of 66 mm min^−1^ within one driving cycle (Figure 13f). However, it should be noted that the crawling speed of this humidity-responsive actuator is much slower than that of light-responsive actuators. Hence, there remains an urgent need for highly robust and fast-response humidity-driven crawling actuators. The actuator, with dimensions of 9 mm × 1 mm, starts off in a partially overlapping state. Upon application of ample moisture, it tends to transition from curled to straightened. As energy continuously accumulates within the actuator, it eventually jumps up through sudden energy release. The jumping film can reach a height of approximately 5.3 mm within 0.04 s, followed by a descent onto the substrate in 0.08 s and returning to its original curled state. This allows for continuous jumping ability and maintains functionality when exposed to high humidity conditions again.

Compared with the light-responsive jumping actuator, the jumping height is relatively low. Lu et al. [164] proposed a method of constructing a multiresponsive gel actuator by adsorbing the MG-CMA gel layer onto a non-responsive PAAm and then performing photocrosslinking. The bending diagram under different conditions is shown in their study (Figure 13g). At pH 7.4, the actuator is concave at all temperatures due to the high Q value of the MG gel layer. At pH 6.0, the actuator changes from concave at 10 °C to convex at 50 °C because the Q value is temperature-dependent at this pH. At pH 5.4, the temperature-triggered change is the largest because the Q has the strongest dependence on temperature. Supporting these observations, it can be noted that there is a significant bending angle change of up to 167° at pH 5.4, while it changes by less than 30° under other conditions. This potential useful design feature for future PRLDA actuators demonstrates their applicability. Additionally, a gripper was constructed whereby increasing the pH from 4.5 to 7.4 leads to release of load (an O-ring, 85 mg) underwater as depicted (Figure 13g). The grasping and releasing behaviors are demonstrated which also shows that the bilayer structure exhibits excellent reversibility for temperature-triggered actuation.

In conclusion, these studies have made important progress in the field of hydrogel grippers. Table 1 summarizes the materials, fabrication strategies, advantages, and challenges of hydrogel grippers in the aforementioned applications. Through in-depth research and innovative applications of hydrogels, more possibilities are provided for their development in different fields. In the future, we can expect further improvements in the performance and functionality of hydrogel grippers, such as improving response speed, enhancing stability, and accuracy. It is anticipated that this comprehensive framework will stimulate increased interest among researchers to delve into this field.

**Table 1 biomimetics-09-00585-t001:** Summary of typical research of hydrogel grippers.

Response Type	Materials	Fabrication	Gripper Effect	Advantage	Disadvantages	Ref.
Thermal	PNIPAm, PAA-Ca(CH_3_COO)_2_, P(MAAm-*co*-MAAc) and PNI-PAM	One-step method	It can lift an object 500 times its own weight.	Self-locking ability, high grasping ability, and versatility.	Limited mechanical properties, temperature dependence, and complex preparation.	[103]
	PNIPAm, PAA-Ca(CH_3_COO)_2_, P(MAAm-*co*-MAAc and PNI-PAM	One-step polymerization method	It can withstand a weight more than 47.6 times its own weight.	High load-bearing capacity, multiple bending states, and brush-like adhesion between hydrogels.	The preparation process is complex and relies on temperature control. There may be durability issues.	[94]
	NIPAM and TEMED	One-step polymerization method	Compared with traditional PNIPAM materials, the weight is reduced by 7.5 times.	Fast response, high power output, reusable, and no chemical modification required.	Sensitive to temperature, dependent on water environment, and has a complex structure.	[115]
	NIPAM, AA, PEGDA	One-step ultraviolet polymerization method	Response performance in water at 60 °C within just 9 s.	Ultrafast thermal response speed, high strength, and biomimicry.	The preparation process is relatively complex, the material cost is high, and it is limited by temperature.	[116]
Photothermal	ANF/GNP and PEG	One-step method	Maintain a temperature above 90 °C when the light power density is 200 mW cm^−2^.	It has a high in-plane thermal conductivity, excellent photothermal conversion performance, and temperature-dependent flexibility and shape memory behavior.	The mechanical properties may be relatively low, and the preparation process is slightly complicated.	[117]
	AAm, NIPAAm, TA, and PDMS	One-step method	It can be increased from 17.9 °C to 107 °C within 30 s.	Highly biomimetic, near-infrared responsive, reversible conversion, and good flexibility.	Dependence on near-infrared light, limited temperature range, and limited load-bearing capacity.	[98]
Electric	VBS, APS, CaCl_2_, and TMEDA, AAm, HEMA, PBS, FBS, and CCK	Ultraviolet light irradiation process method	Even at a low voltage of 10 V, it exhibits a bending deflection of more than 100° within 1 min.	Rapid deformation at low voltage, good biocompatibility.	Limited mechanical strength and high energy consumption.	[124]
	TREN, PAA, TREN solution	Ultraviolet polymerization	Capable of gripping and lifting objects weighing approximately 31 times the weight of the gripper.	High performance, low power consumption, high strain capacity, long life, high energy density, and multi-degree-of-freedom motion	Copper pole limitations affect bending performance, and its widespread use is dependent on external support.	[125]
Magnetic	MAAm, Fe_3_O_4_ and KPS, MAAc, PVA, TMEDA, MBAA, NH_4_OH, MPS, TEOS	One-step polymerization method	Excellent mechanical properties, up to 19.7 MPa Young’s modulus, 14.6 MPa tensile breaking stress, and 390% strain at break.	Outstanding reversible drive deformation capability, its stiffness and shape can be precisely controlled by adjusting temperature and magnetic field strength.	Requires an alternating magnetic field as an external stimulus to drive the deformation of the hydrogel, which does not allow for fully autonomous actuation.	[129]
	GelMa, LAP, SPIONs	Ultraviolet polymerization	Can lift up to 9.5 mg at a magnetic field strength of 5–25 mT.	Programmable 3D magnetic anisotropy, biodegradability, multifunctionality.	Limited load capacity, more complex manufacturing process, magnetic field dependence.	[48]
Electric–magnetic	CAA, PEGA, DMC, AM, MBA, MeOH, PBS, TPO	Ultraviolet polymerization	Under lower electric fields (2 V cm^−1^ and 3 V cm^−1^), the time required for the application of an electric field to a hydrogel to reach the maximum bending angle is approximately 120 s. While under higher electric fields (4 V cm^−1^ and 5 V cm^−1^), it is around 80 s.	Capable of performing simultaneous gripping and moving of objects, it can be manipulated remotely and is biocompatible.	Electrode fragmentation issues, reliance on clean electrolytes, and optimization of electrode materials.	[132]
	AlgNa, PVA, acrylamide gel, CaCl_2_	Microfloppies control technology	An ultrafast response of 260.5 mm s^−1^ with high-frequency controllability (6 Hz) and a large deformation of 172% with hydrogel actuation are observed.	Ultrafast response, high-frequency control, handling of fragile objects	Moisture loss due to heat generated by current flow.	[133]
pH	AA and DMAEMA	One-step method	The soft clip bends gradually in water, holding the copper block and lifting it by 2 mm (hydrogel clip 10.2039 g, a piece of copper 2.2921 g).	PAD4 hydrogel showed complex deformation under different pH conditions and recovered to its original shape in ethanol. High mechanical strength.	The tensile fracture strength decreases after expansion. The response speed slows down. The bending angle decreases.	[142]
pH and thermal dual-responsive	P(AAm-AAc-3-AAPBA)/PNIPAM	Ultraviolet polymerization	It takes about 70 s to change the bending angle of the bilayer structure from 355 (10 °C, pH 2) to 360 (55 °C, pH 12).	Very sensitive to the temperature of the surrounding environment. The bending angle is large. The bending speed is fast. It can be reused.	The preparation process is complicated. The mechanical strength is low.	[143]
	CS hydrogel	One-step polymerization method	The flexibility is improved in sodium chloride solution (elongation at break is 43.40 ± 3.46% and Young’s modulus is 133.29 ± 24.61 kPa).	Fast response. High power output. Fast bending speed.	After swelling in deionized water, it becomes rigid and the elongation at break decreases. Dependent on the water environment.	[144]
Ionic strength and thermal dual-responsive	Alg-PDMAEMA layer	Crosslinking synthesis method	Complex deformation from 2D to 3D can be realized.	Complex bending lines can be realized.	Dependence on water environment. Low mechanical strength.	[145]
Electrochemistry	Chitosan hydrogel	One-step method	The gripper can bend automatically in about 30 s, and its bending response is faster.	Fast reaction time. Shape memory effect. Reusable.	Dependence on electrolyte solution. Complex driving environment.	[146]
Light	MG-CMA	UV-light, PRLDA	The groups enable light to directly assemble gels and adjust mechanical and swelling properties without the use of small molecules or free radical polymerization.	Enables light to directly assemble gels and adjust mechanical and swelling properties without the use of small molecules or radical polymerization.		[164]
DNA	DNA molecules	One-step method	The high degree of swelling of the DNA gel can lead to bending of structures that are a millimeter to a centimeter thick. First, the bilayer is 10 mm long × 7.23 mm thick, with a maximum expansion ratio of 3.72 ± 0.11, and should be folded into a complete circle after sequence-specific DNA trigger drive.	The gel is able to respond to specific DNA trigger signals, enabling complex and programmable shape changes.	Multistage, goal-oriented behavior that is not currently achievable.	[117]
Ultrasound	NIPAM	3D printed	The printability of AAm-based inks is between 0.21 mm and 0.41 mm nozzle diameters, and the corresponding printing pressure is between 15 and 45 kPa. In the case of NIPAM-based inks, a precise range of printing pressures (10–30 kPa) is observed in the range of nozzle diameters from 0.21 to 0.41 mm.	Compliant.	The attenuation coefficient is large.	[165]
Moisture	PAM	Crosslinking	At 60% ΔRH, the bending angle of the actuator with different PET thicknesses varies. The 22 μm thick PET actuator has a response time of 6 s and a recovery time of 9 s, with a maximum bending angle of approximately 297°.	With a small temperature change (3.9 °C), OS oscillation drives with a large oscillation amplitude (14.4 mm) can be realized.	Creeping actuators: much slower than light-responsive actuators. Moisture control electric switch: the frequency is relatively slow. Jump actuators: light-responsive jump actuators have a relatively low jump height. Mechanical gripper: the gripping time and weight ratio of the gripper cannot meet the requirements.	[162]

## 5. Conclusions and Perspectives

Soft hydrogel grippers have significantly advanced in the healthcare, soft robotics, industry, etc. fields through their stimuli-responsive grasping and releasing ability. This work delves into various hydrogel types (e.g., thermal/electrical/magnetic/chemical/etc.) which can be combined with other materials to fabricate soft hydrogel grippers, and the manufacturing process, including one-step polymerization, 3D/4D printing, and structural modification, is summarized. Particular attention is paid to recent advancements in actuating mechanisms of soft hydrogel grippers, focusing on thermal, magnetic, electric, and chemical aspects, ending with the grabbing of various response object by soft hydrogel grippers in emerging applications. Although significant progress has been made in the field, there is room for the development of a new generation of soft hydrogel grippers (Figure 14).

First, there is a notable direct influence in comprehending the mechanical properties, stimuli-response mechanisms, dehydration resistance, and cost stemming from the selection of hydrogel monomer, crosslinking agent, curing agent, addition elements, and the design of structure. Therefore, optimizing material selection and structural design is the core key to building high-quality hydrogel grippers. At present, the phenomenon of the hydrogel gripper’s lagging response to external conditions also needs to be addressed from the abovementioned aspects. Meanwhile, stability, durability, and robustness under extreme environmental conditions may require further research and improvement. For instance, in the electrically responsive system, long-term exposure to high voltage and prolonged application give rise to the presence of debris around the electrodes, which impairs the response to the external environment. The use of clean electrolytes will raise the cost of the experiment as well. Additionally, hydrogel grippers face issues such as short life span caused by wear and tear and specific stimuli-response conditions, which means they have limited applicability.

Second, to maintain the hydrogel grippers’ ability to respond to external conditions, energy needs to be continuously consumed, resulting in waste. To solve the problem, adding energy circulation systems may increase efficiency and save energy. However, complex supporting parts make hydrogel gripper systems huge, and hydrogel gripper systems tend to be small-scale integrated systems except for special situations in the future. Additionally, there is an urgent need for a reliable and cost-effective manufacturing process of minimized hydrogel grippers. It is worth mentioning that 3D or 4D printing pave the way to fabricating elaborate structures of hydrogel grippers, but these technologies have high requirements on the properties of printing materials. Therefore, to achieve the customization and refinement of the structure through manufacturing additive technology, the development of hydrogel materials is required. On the other hand, due to the limitations of hydrogel dehydration and poor mechanical properties, it is usually necessary to compound hydrogels with other materials to fabricate grippers. The degree of bonding at heterogeneous interfaces will affect the mechanical properties. Therefore, an in-depth study of the bonding mechanism of heterogeneous interfaces is also of great importance.

Last, but not least, the applications of hydrogel grippers stay in the imagination stage, and it is necessary to continue work at the level of practical applications. For example, a 4D printed hydrogel microgripper has a good application prospect in the biomedical field, but its degradability and safety need to be further researched. Fortunately, recent advancements in various hydrogel materials offer promising prospects for the development of various characteristics of applications. In addition, multiresponse coupling is also a development trend due to expand the various application scenarios.

In summary, the hydrogel gripper is an emerging area, which is assuming a potentially pivotal role across various domains including soft robotics, healthcare, sensors, etc. [166]. Hydrogel grippers have been tentatively applied on a small-scale commercial basis in the biomedical and food-processing fields. Small enterprises, in cooperation with scientific research institutions, promote the initial industrialization. However, due to high costs and low efficiency, the scale is small. They face challenges in terms of cost, technology, and competition during the commercialization/industrialization process. Nevertheless, with the development of materials science and manufacturing technology, breakthrough key indicators and the realization of the transformation towards large-scale industrial production are expected. This work provides a comprehensive overview of materials, manufacturing process, stimuli-response mechanisms, applications, challenges, and perspectives of hydrogel grippers, providing useful information for the development of a new generation of hydrogel grippers. In addition, to enhance the performance and reduce the duction costs of hydrogel grippers, we believe that a solid foundation of knowledge and interdisciplinary collaboration are essential.

## Figures and Tables

**Figure 1 biomimetics-09-00585-f001:**
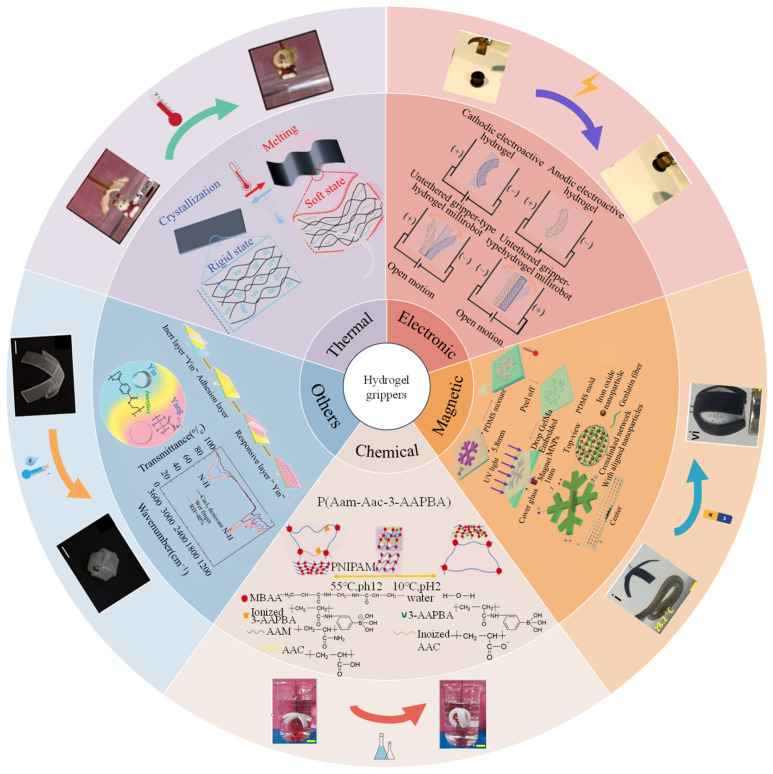
Illustration of various stimuli-responsive hydrogel grippers.

**Figure 2 biomimetics-09-00585-f002:**
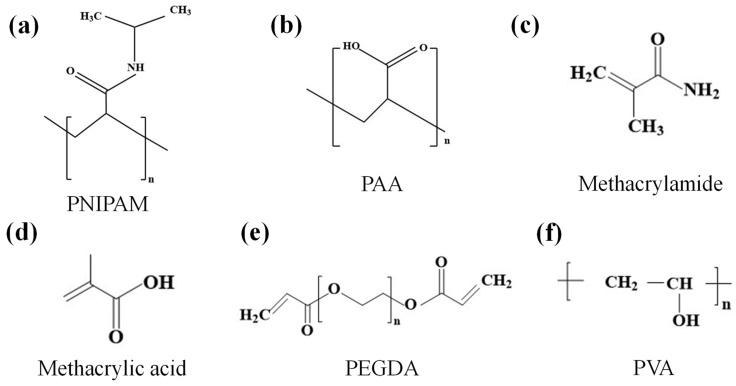
The molecular structure of hydrogels (**a**) PNIPAM, (**b**) PAA, (**c**) Methacrylamide, (**d**) Methacrylic acid, (**e**) PEGDA, (**f**) PVA.

**Figure 3 biomimetics-09-00585-f003:**
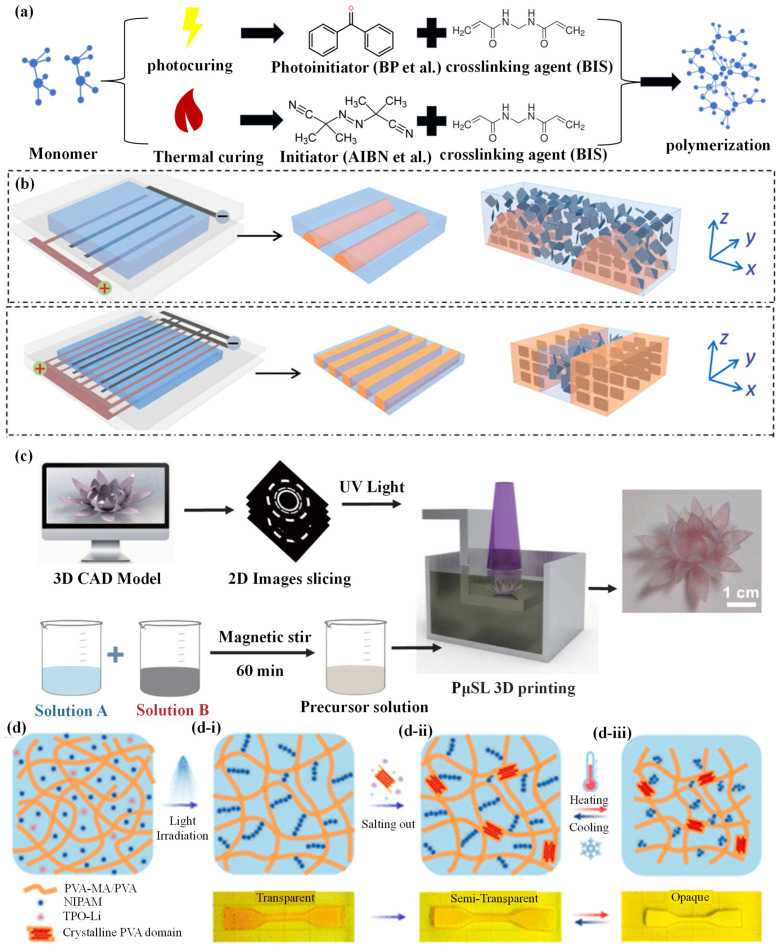
(**a**) Schematic illustration of the one-step preparation process of hydrogel. (**b**) Schematic for the fabrication of hydrogel with a through-thickness gradient and in-plane gradient by employing electrodes patterned on one substrate for the electric orientation of nanosheets. The other substrate of the reaction cell lacks electrodes [74]. Copyright 2021, Wiley-VCH GmbH. (**c**) The specific synthesis process of the NIPAAm-based hydrogel via the PµSL-based 3D printing technique [75]. Copyright 2021, IOP Publishing Ltd. on behalf of the IMMT. (**d**) The aqueous precursor consisting of PVA, PVA-MA, NIPAM, and TPO-Li. (**d-i**) The one-pot synthesis of PVA/(PVA-MA)-g-PNIPAM hydrogel through light irradiation from a DLP 3D printer. The hydrogel that was just printed was transparent. (**d-ii**) The toughening of the hydrogel by immersing it in Na_2_SO_4_ salt solution to induce the aggregation and crystallization of PVA. After the salting-out process, the hydrogel became semitransparent. (**d-iii**) The actuation of the hydrogel by heating, and the recovery of the hydrogel by cooling. After heating, the hydrogel turned completely opaque and reverted to semitransparent after cooling [76]. Copyright 2020, American Chemical Society.

**Figure 5 biomimetics-09-00585-f005:**
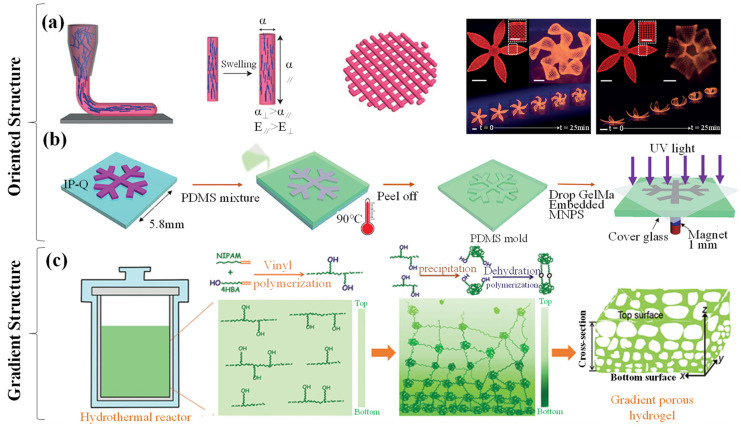
(**a**) Bio-4D printing of flower with cellulose fibrils [106]. Copyright 2016, Springer Nature. (**b**) Manufacturing solutions for milli-gripper [48]. Copyright 2016, Wiley-VCH Verlag. (**c**) Hydrogels with gradient/different distributions in thickness [104]. Copyright 2018 Wiley.

**Figure 7 biomimetics-09-00585-f007:**
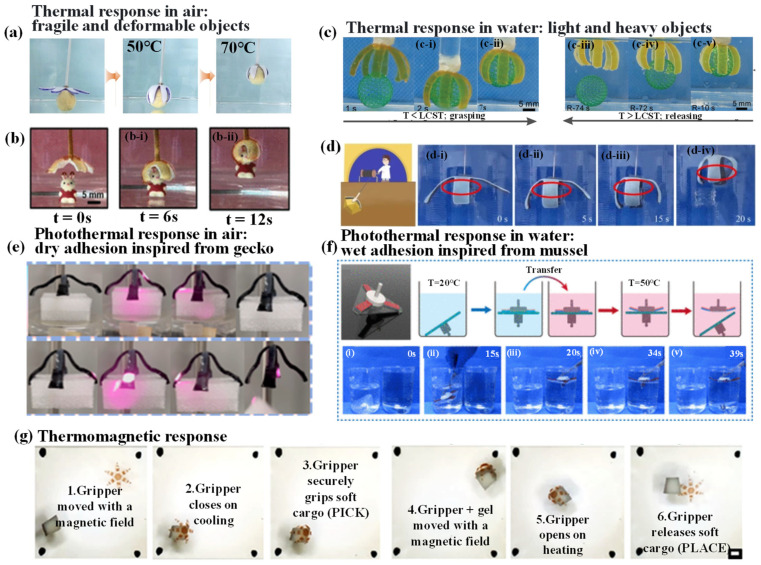
(**a**) A hydrogel gripper deforms and grabs the yolk at 50 °C and lifts the yolk after stiffening at 70 °C [103]. Copyright 2024, Science China Press. (**b**) Schematic illustrations of integrated gradient hydrogels as thermoresponsive grippers that can be used to grab a Miffy toy (4 °C to 60 °C) [116]. Copyright 2022, American Chemical Society. (**c**) Digital photos of the thermal-responsive gripper to grasp and release a hollow ball [47]. Copyright 2019, WILEY-VCH Verlag GmbH & Co. KGaA, Weinheim. (**d**) Process of PSM hydrogel gripper grabbing metal sheet from 50 °C water [118]. Copyright 2024, the author(s). The adjusted four-arm gripper (**e**) grasping an object and releasing the object [98]. Copyright 2022, Elsevier Ltd. (**f**) A schematic diagram of a gecko’s feet-like SPSA device, which is firmly attached to the surface of triangular PMMA plastic with 12 g of cargo and can be used to transfer objects from a cold water bath of 20 °C to a hot water bath of 50 °C. Among them, (**f**-**i**–**f**-**v**) are the physical diagrams of this process [119]. Copyright 2021, American Chemical Society. (**g**) Demonstration of the thermomagnetic response operation of a soft gripper. Sequential optical images of the magnetically guided and thermally driven soft gripper are obtained when using a permanent magnet and a hot plate for picking up and placing soft cargoes, and no visible damage occurs [120]. Copyright 2018, American Chemical Society.

**Figure 9 biomimetics-09-00585-f009:**
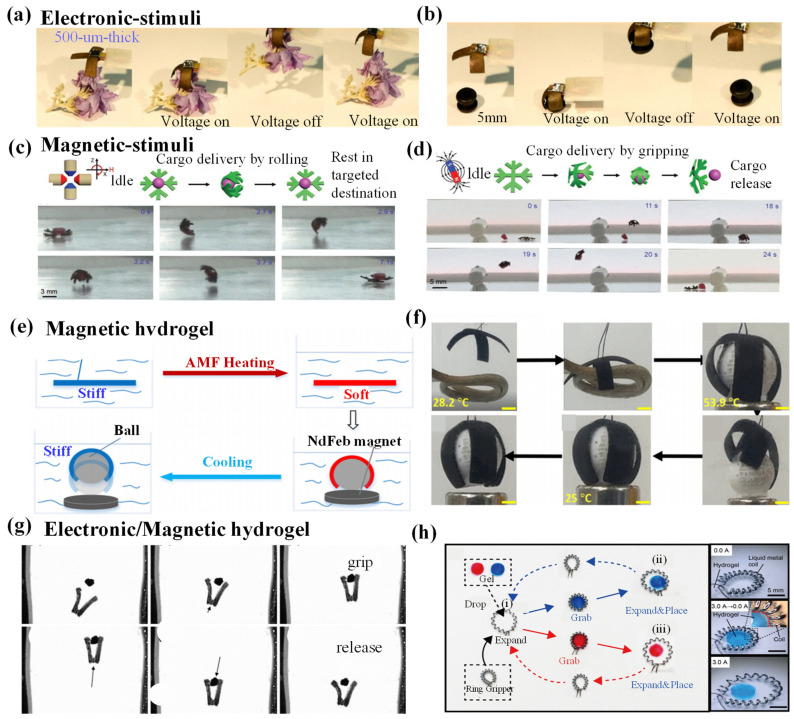
(**a**) A flexible fixture with a flat structure (500 µm thick) holds 20 mg of flowers. (**b**) The clamp holds a 1000 mg battery [125]. Copyright 2020, American Chemical Society. (**c**) Snapshots of rolling transported cargo in a rotating magnetic field of 25 mt. (**d**) Gripping of the cargo over obstacles under the gradient of the magnetic field generated by the permanent magnets to complete the cargo transporting and release task [48]. Copyright 2020, the authors. (**e**) A schematic diagram illustrating the design of a magnetic soft hydrogel underwater gripper. (**f**) A plastic ball weighing 3.75 g is lifted by a four-arm magnetic soft hydrogel underwater gripper (weighing 0.52 g) [129]. Copyright 2024, the Royal Society of Chemistry. (**g**) The hydrogel microrobot approaches the object under the guidance of the magnetic field, closes and grabs when the electric field is activated, and moves with the magnetic field. The reversal of the electric field releases the object for precise manipulation [132]. Copyright 2020, IOP Publishing Ltd. (**h**) Demonstration of grabbing, moving, and releasing gel objects using ring gripper [133]. Copyright 2022, the authors.

**Figure 10 biomimetics-09-00585-f010:**
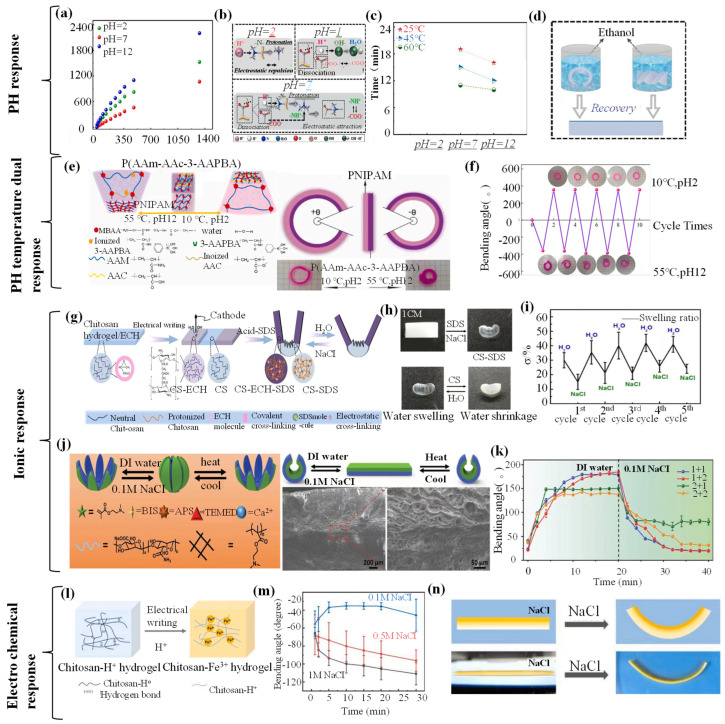
(**a**) The extent of swelling of PAD4 in acidic solution, deionized water, and alkaline solution. (**b**) The swelling mechanism has been researched in acidic solution. (**c**) PAD4 with small specific surface area in acidic, neutral, and alkaline medium at 25 °C, 45 °C, 60 °C had its bending time determined. (**d**) The PAD4 (with a width of 6 mm and a thickness of 2 mm) underwent a recovery process in ethanol. The scale indicators were marked at 10 mm [142], Copyright 2022. Elsevier B.V. (**e**) Schematic of the P(AAm-AAc-3-AAPBA)/PNIPAM bilayer’s bending response to varying conditions: 55 °C, pH 12, and 10 °C, pH 2. (**f**) Reversible bending of P(AAm-AAc-3-AAPBA)/PNIPAM bilayers in acidic (10 °C, pH 2) and alkaline (55 °C, pH 12) environments [143], Copyright 2023, Published by Elsevier Ltd. (**g**) Regulating CS hydrogel crosslinking through electrical writing yields actuators with shape memory. (**h**) Illustration of CS−SDS hydrogel formation and its shape memory capability. (**i**) CS−SDS hydrogel swelling ratio in alternating deionized water and NaCl solution [144], Copyright 2022, American Chemical Society. (**j**) Schematic of a dual-responsive, hollow spherical hydrogel actuator with pH and ionic strength sensitivity, featuring illustrations of its ionic strength and thermal actuation, and a cross-sectional SEM image of the bilayer hydrogel. (**k**) Reversible actuation of bilayer hydrogels with varying thicknesses induced by ionic strength [145], Copyright 2020, WILEY-VCH Verlag GmbH & Co. (**l**) Fe^3+^-complexed chitosan gradient hydrogel’s deformation response to various solutions, prepared via anodic electrical writing. (**m**) Bending angle variation over time in NaCl solutions of varying concentrations. (**n**) Actuator’s reverse bending in deionized water and NaCl solution [146], Copyright 2021, Elsevier Ltd.

**Figure 12 biomimetics-09-00585-f012:**
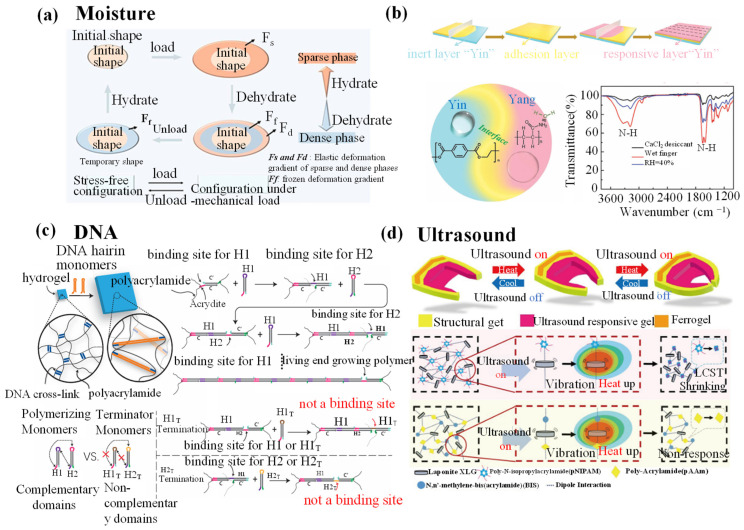
(**a**) The schematic of water-triggered hydrogel process [161]. Copyright 2023, American Society of Mechanical Engineers. (**b**) The composition of YYI actuators [162]. Copyright 2023, Wiley-VCH GmbH. (**c**) DNA-directed expansion of DNA-crosslinked hydrogel [163]. Copyright 2017, American Association for the Advancement of Science. (**d**) The mechanism of ultrasound response hydrogel gripper [94]. 2020, Copyright American Chemical Society.

**Figure 13 biomimetics-09-00585-f013:**
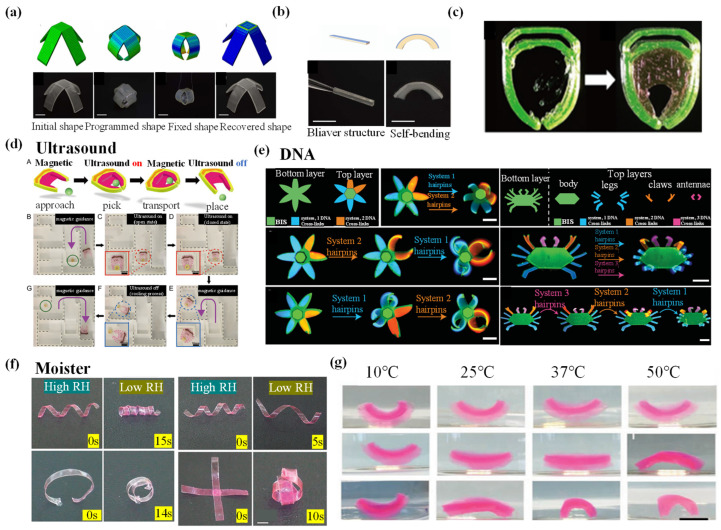
(**a**) Simulated and experimental results of a gripper structure during the shape memory cycle for water-triggered SMH: simulated results and experimental results [161]. Copyright 2023, by ASME. (**b**) Comparison of simulation and experimental results of a self-bending bilayer structure [161]. Copyright 2023, by ASME. (**c**) Figures illustrating the progression of 3D printed hybrid gripper from non-responsive structural stimuli gel to ultrasound responsive gel [165]. Copyright 2020, American Chemical Society. (**d**) Pick-and-place task of the hybrid gripper involves a series of steps. Schematics illustrate the process of pick-and-place task inside a maze, while experimental snapshots provide visual documentation. The hybrid gripper, initially located at a specific point, approaches the target using magnetic guidance. It then picks up the target by absorbing ultrasounds (on-mode), transports it using magnetic guidance, places it by desorbing ultrasounds (off-mode), and finally exits the maze again guided by magnets. All scale bars in the illustrations represent 10 mm [165]. Copyright 2020, American Chemical Society. (**e**) DNA-sequence-programmed shape change of macroscopic hydrogel shapes. The figure shows a schematic of a six-petal flower, where all petals curl in response to both system 1 and 2 hairpins. Specific petals actuate in response to either system 1 or system 2 hairpins alone and can be actuated in series. Additionally, the hydrogel crab schematic demonstrates that legs, claws, and antennae all actuate in response to system 1, 2, and 3 hairpins through serial actuation. The solutions contained 20 mM of each hairpin with 98% polymerizing hairpins and 2% terminating hairpins. DNA-crosslinked hydrogel domains are differentially colored for clarity [163]. Copyright 2017, The authors, some rights reserved; exclusive licensee American Association for the Advancement of Science. (**f**) Programmable morphing of the YYI actuator under high and low humidity is achieved through the following mechanisms: a left-handed coil actuator with the PAM hydrogel on the inside, a right-handed coil actuator with the PAM hydrogel on the outside, a rolled actuator, and an artificial flower with opened and closed states [162]. Copyright 2023, Wiley-VCH GmbH. (**g**) Photographs of the bilayer gel were taken at pH 5.4, 6.0, and 7.4, and temperatures ranging from 10 to 50 °C [164]. Copyright 2020, the authors. Published by WILEY-VCH Verlag GmbH & Co. KGaA, Weinheim.

**Figure 14 biomimetics-09-00585-f014:**
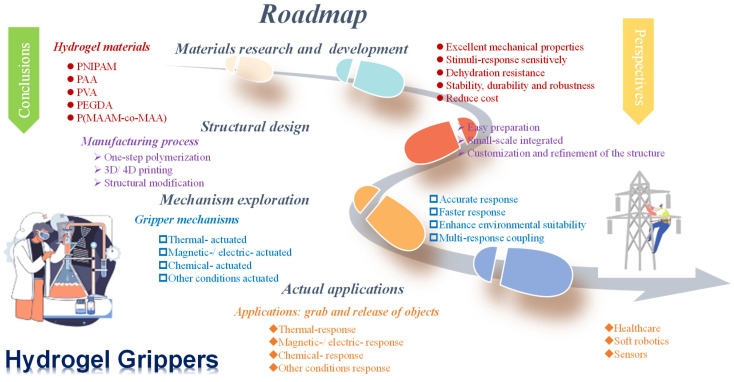
Conclusions and perspectives of soft hydrogel gripper.

## Data Availability

No primary research results, software, or code has been included and no new data were generated or analyzed as part of this review.

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
