# Peer review of "Advancement in Soft Hydrogel Grippers: Comprehensive Insights into Materials, Fabrication Strategies, Grasping Mechanism, and Applications"

_biomimetics, 2024, doi:10.3390/biomimetics9100585_

Round 1

Reviewer 1 Report

Comments and Suggestions for Authors

This review article provides an extensive and detailed description of materials, fabrication methods, and applications of the soft hydrogel grippers. The amount of description suggests that the paper is well researched and organized. I think this article will be a useful reference for an overview of this topic.

Although I am not able to confirm and evaluate the details, I would like to make the following comments to make the article more useful.

Figure 1 is an illustration of the overall subject of the soft hydrogel grippers. Although it is a schematic diagram, the chemical formula and some other details are unclear. It would be helpful if the illustration could be corrected to be clearer.

Figures 2 and 3 are not consistent with references in the text. Please correct them.

2.3 PEDGA  Is this a misnomer for PEGDA?

Also, please describe the spelled out terms where the term is first mentioned.

The table 1 shows a comparison of the various materials.

Since it is difficult to understand the differences in characteristics from the table alone, if possible, it would be better to have a graph comparing important characteristics of the materials such as deformation, output, response speed, etc.

Comments on the Quality of English Language

I think the English expression is fine.

Author Response

This review article provides an extensive and detailed description of materials, fabrication methods, and applications of the soft hydrogel grippers. The amount of description suggests that the paper is well researched and organized. I think this article will be a useful reference for an overview of this topic.

1.Figure 1 is an illustration of the overall subject of the soft hydrogel grippers. Although it is a schematic diagram, the chemical formula and some other details are unclear. It would be helpful if the illustration could be corrected to be clearer.

Reply: Thank you for your valuable comments. We have made changed in Figure 1 as following.

2.Figures 2 and 3 are not consistent with references in the text. Please correct them.

Reply: Thank you for your comments. We have corrected the literature in Figures 3. Figure 2 does not involve literature.

3.2.3 PEDGA Is this a misnomer for PEGDA?

Also, please describe the spelled out terms where the term is first mentioned.

Reply: Thank you for your comments. Here is a spelling error, we have corrected PEDGA to PEGDA. In addition, we will provide all spelling terms for the first mentions of the full text.

4.The table 1 shows a comparison of the various materials.

Since it is difficult to understand the differences in characteristics from the table alone, if possible, it would be better to have a graph comparing important characteristics of the materials such as deformation, output, response speed, etc.

Reply: Thank you for your comments. Our review mainly elaborates from three aspects: thermal response, magnetic response, and chemical response. The materials involved in each response are different. Table 1 has summarized the current different types of hydrogel response methods of this kind. For each article, the materials used are different, and each article has its own characteristics. Our discussion has already reflected the respective deformation output response characteristics of each article, that is, the content in the "Gripper effect" column in Table 1. Since each article has certain differences, we have presented it directly in this form in the discussion.

Reviewer 2 Report

Comments and Suggestions for Authors

Li and coworkers reviewed the development of responsive hydrogel based gripper. The manuscript covers the material chemistry, manufacturing strategies, and driving mechanisms. The reviewer thinks the review is comprehensive and well-organized. The related reference is up-to-date. Publication is suggested.

However, a few finer points can be improved:

1)The reviewer suggest adding one paragraph in the introduction section to introduce the structure of this review so the readers can better access sections of their interest.

2)It is of the authors' choice, but it is of reviewer's interest to learn about the progress in commercialization/industrialization of hydrogel-based grippers.

3)It is suggested to briefly talk about the advantages/limitation of hydrogel materials compared to other popular soft responsive materials that are used to make grippers, such as shape memory polymer, dielectric elastomer, liquid crystal elastomer, etc.

Author Response

Li and coworkers reviewed the development of responsive hydrogel based gripper. The manuscript covers the material chemistry, manufacturing strategies, and driving mechanisms. The reviewer thinks the review is comprehensive and well-organized. The related reference is up-to-date. Publication is suggested.

However, a few finer points can be improved:

1.The reviewer suggest adding one paragraph in the introduction section to introduce the structure of this review so the readers can better access sections of their interest.

Reply: Thank you for your comments. We have made some revisions and written the newly - revised paragraphs at the last paragraph of the first chapter of the article.

The following is the content we added: This review focuses on stimuli-responsive soft hydrogel grippers. It aims to explore the material composition and characteristics of stimuli-responsive hydrogels, and also to explore the relevant efforts in combining hydrogels with other materials to fabricate various such grippers (Figure 1). Specifically, the introduction part expounds the research background of soft grippers and their application potential in multiple fields. The materials part elaborates on the molecular structures, functional groups, characteristics of common hydrogels and their influence on the actuation of hydrogel grippers. The fabrication strategies part includes one-step synthesis of direct polymerization and fabrication additives, as well as structural modification strategies such as bilayer, patterned, multilayer, oriented and gradient structures, and also involves doping strategies, etc. The stimuli-responsive hydrogel grippers part discusses hydrogel grippers under external stimuli-responsive mechanisms such as thermal, electrical/magnetic, and chemical stimuli, explores other response types and expounds on multi-responsive hydrogel grippers. Reviewing the latest progress in this field to introduce the applications of soft hydrogel grippers is also a key content of this review. Finally, the conclusion and outlook part summarizes the research progress, points out the existing problems, and presents an outlook for future development.

2.It is of the authors' choice, but it is of reviewer's interest to learn about the progress in commercialization/industrialization of hydrogel-based grippers.

Reply: Thank you for your comments. We investigated and added the progress in the commercialization/industrialization of hydrogel-containing fixtures. And we incorporated the research findings into the fifth paragraph of the first chapter and the last paragraph of the fifth chapter of the article.

The following is the content we added: In the early stages of commercialization and industrialization, hydrogel grippers were mostly in the laboratory research phase. Researchers explored their basic properties and attempted to apply them to gripper design. Although they recognized the potential advantages, it was difficult to achieve commercial applications due to technological limitations. In order to meet production requirements, researchers made efforts in two aspects: material improvement (adding nanomaterials, etc. to optimize performance and adopting new methods to increase speed) and prototype development (using optimized materials to develop simple prototypes and having initial feasibility in biomedical research).

Hydrogel grippers have been tentatively applied on a small-scale commercial basis in the biomedical and food-processing fields. Small enterprises, in cooperation with scientific research institutions, promote the initial industrialization. However, due to high costs and low efficiency, the scale is small. They face challenges in terms of cost, technology and competition during the commercialization/industrialization process. Nevertheless, with the development of materials science and manufacturing technology, it is expected to break through key indicators and realize the transformation towards large-scale industrial production.

3.It is suggested to briefly talk about the advantages/limitation of hydrogel materials compared to other popular soft responsive materials that are used to make grippers, such as shape memory polymer, dielectric elastomer, liquid crystal elastomer, etc.

Reply: We really appreciate the reviewer’s comment. We have carefully investigated the advantages/limitations of hydrogel materials in comparison with other popular soft - responsive materials (such as shape - memory polymers, dielectric elastomers, liquid - crystal elastomers, etc.) used for fabricating grippers, and written the relevant text in the second paragraph of the first chapter of the article.

The following is the content we added: Compared with soft responsive materials often used in manufacturing fixtures, such as shape-memory polymers, dielectric elastomers, and liquid-crystal elastomers, hydrogels have advantages. They have good biocompatibility, excellent softness, and diverse environmental responsiveness. However, they also have limitations. Their mechanical properties are low, and they are prone to damage under large external forces. Their response speed is slow, and it is difficult to adjust in scenarios requiring rapid response. Moreover, they contain a large amount of water, and their structure and performance change significantly or even become inactivated when dried. In contrast, other materials have advantages in terms of mechanical properties, response speed, and stable performance in dry environments.
